EMBO
Molecular Medicine

# LSD1 inhibition induces differentiation and cell death in Merkel cell carcinoma

Lukas Leiendecker[1,‡] (ID), Pauline S Jung[1,2,‡] (ID), Izabela Krecioch[1] (ID), Tobias Neumann[1] (ID), Alexander Schleiffer[1] (ID), Karl Mechtler[3] (ID), Thomas Wiesner[2,*] (ID) & Anna C Obenauf[1,**] (ID)

## Abstract

**Merkel cell carcinoma (MCC) is a highly aggressive, neuroendocrine skin cancer that lacks actionable mutations, which could be utilized for targeted therapies. Epigenetic regulators governing cell identity may represent unexplored therapeutic entry points. Here, we targeted epigenetic regulators in a pharmacological screen and discovered that the lysine-specific histone demethylase 1A (LSD1/KDM1A) is required for MCC growth *in vitro* and *in vivo*. We show that LSD1 inhibition in MCC disrupts the LSD1-CoREST complex leading to displacement and degradation of HMG20B (BRAF35), a poorly characterized complex member that is essential for MCC proliferation. Inhibition of LSD1 causes derepression of transcriptional master regulators of the neuronal lineage, activates a gene expression signature resembling normal Merkel cells, and induces cell cycle arrest and cell death. Our study unveils the importance of LSD1 for maintaining cellular plasticity and proliferation in MCC. There is also growing evidence that cancer cells exploit cellular plasticity and dedifferentiation programs to evade destruction by the immune system. The combination of LSD1 inhibitors with checkpoint inhibitors may thus represent a promising treatment strategy for MCC patients.**

**Keywords** epigenetics; HMG20B; LSD1; merkel cell carcinoma; targeted therapy

**Subject Categories** Cancer; Chromatin, Transcription & Genomics; Skin

See also: **F Mauri & C Blanpain** (November 2020)

## Introduction

Merkel cell carcinoma (MCC) is an aggressive neuroendocrine carcinoma of the skin that commonly develops in elderly and immuno-suppressed patients. Approximately 80% of MCCs are associated with the clonal integration of the Merkel cell polyomavirus (MCV), whereas the remaining MCCs show a high number of mutations induced by chronic sun exposure (Feng *et al*, 2008; Harms *et al*, 2015). In virus-positive MCC, the viral large T antigen binds and inactivates RB, while the small T antigen interacts with various proteins, including EP400/MYCL, FBXW7, and CDC20 to promote oncogenesis (Harms *et al*, 2018). Whereas virus-positive MCC harbors a low mutational burden and no recurring oncogenic alterations, virus-negative MCC shows a ~ 100-fold higher mutational load with recurrent inactivating mutations in *RB1* and *TP53* (Harms *et al*, 2015). Conventional treatments such as radio- or chemotherapy have limited and short-lived clinical efficacy (Tai *et al*, 2000; Iyer *et al*, 2016). Immunotherapy with checkpoint inhibitors, such as anti-PDL1/-PD1, has proven to be effective for the treatment of MCC; however, only about half of the MCC patients respond to immunotherapy, highlighting the need for novel therapeutic entry points (Kaufman *et al*, 2016; Nghiem *et al*, 2016).

Epigenetic regulators govern cell identity and differentiation, and their dysregulation or mutation can give rise to aberrant cell states and cancer (Shen & Laird, 2013; Flavahan *et al*, 2017). To overcome differentiation blockades in cancer cells, epigenetic regulators have emerged as accessible entry points for targeted therapies (Jones *et al*, 2016; Kelly & Issa, 2017). Given the lack of actionable mutations in MCC, we performed a screen targeting epigenetic regulators to identify potential therapeutic vulnerabilities. We identified the lysine-specific histone demethylase 1A (LSD1/KDM1A) as a strong genetic and pharmacological dependency in MCC. LSD1 maintains pluripotency, represses developmental programs by removing mono- and di-methylation marks on histone H3 lysine 4 (H3K4), and is overexpressed in various cancer types (Andrés *et al*, 1999; Wang *et al*, 2007; Adamo *et al*, 2011). In neuronal tissues, an LSD1 splice isoform, LSD1 + 8a, that demethylates histone H3 lysine 9 (H3K9), has been described (Laurent *et al*, 2015).

Here, we show that pharmacologic inhibition of LSD1 in MCC reduces cell growth and promotes cell death *in vitro* and *in vivo*. LSD1 inhibition derepresses key regulators of the neuronal lineage and impairs the integrity of the LSD1-CoREST complex leading to degradation of HMG20B, an essential subunit of this complex. Our

1 Research Institute of Molecular Pathology (IMP), Vienna BioCenter (VBC), Vienna, Austria
2 Department of Dermatology, Medical University of Vienna, Vienna, Austria
3 Institute of Molecular Biotechnology (IMBA), Vienna BioCenter (VBC), Vienna, Austria
 *Corresponding author. Tel: +43 1 40400 77020; E-mail: thomas.wiesner@meduniwien.ac.at
 **Corresponding author. Tel: +43 1 79730 3060; E-mail: anna.obenauf@imp.ac.at
 ‡These authors contributed equally to this work

results provide a rationale for evaluating LSD1 inhibitors, which are in clinical trials for patients with haematopoietic and solid cancers (Fang et al, 2019), in MCC.

# Results

## LSD1 is required for Merkel cell carcinoma proliferation *in vitro*

To assess the therapeutic potential of epigenetic regulators in MCCs, we performed a pharmacological screen with 43 compounds targeting epigenetic regulators in the virus-positive MCC cell lines PeTa, MKL-1, WaGa, and MS-1 and used human dermal fibroblasts (HDFB) as a control cell line. Compounds targeting EP300/CBP, BRD1/TAF1, and BET family proteins reduced cell growth of MCC but also of HDFBs, indicating a low specificity and general toxicity. In contrast, the LSD1 inhibitor (LSD1i) GSK-LSD1 strongly reduced the growth of all MCC cell lines, but not of HDFBs (Fig 1A). To confirm these results, we treated cells with the structurally related LSD1i ORY-1001 (Maes et al, 2018) and also observed a specific inhibition of MCC cell growth (Figs 1B and EV1A). For both, GSK-LSD1 and ORY-1001, the IC50 values were in the low nM range, indicating a high sensitivity of MCC to LSD1i (Fig 1C).

To verify that the effects of LSD1i treatment are due to specific inhibition of LSD1 and not due to off-target effects of the drug, we used RNAi to deplete LSD1 in MCCs. We engineered MCC cell lines to express doxycycline-inducible shRNAs targeting LSD1 (shLSD1.1 or shLSD1.2), Renilla luciferase (shRenilla, negative control), or the essential ribosomal protein RPS15 (shRPS15, positive control) and added doxycycline (Fig EV1B). The LSD1-targeting shRNAs, but not the two control shRNAs, reduced the LSD1 RNA levels to ~ 15% and the protein levels to ~ 30% (Fig EV1C and D). While the Renilla-targeting control shRNA had no effect, LSD1-targeting shRNAs reduced cell growth in all tested MCC cell lines, comparable to the knockdown of the essential protein RPS15 (Figs 1D and EV1E).

Next, we analyzed independent, genome-wide RNAi screening data from the DepMap project, which includes genetic vulnerability maps of the virus-positive MCC cell lines PeTa, MKL-1, and MKL-2 (Tsherniak et al, 2017). We examined the genes encoding the epigenetic regulators from our initial screen and found that LSD1 scores, together with BRD4, PRMT5, TAF1, and WDR5, among the top 5 dependencies for MCC proliferation (Figs 1E and EV1F). To assess the therapeutic potential of these genetic dependencies, we evaluated the sensitivity and specificity of the BRD4 inhibitor JQ1 (Filippakopoulos et al, 2010), the PRMT5 inhibitor GSK591 (Duncan et al, 2016), the TAF1 inhibitor BAY-299 (Bouché et al, 2017), and the WDR5 inhibitor OICR942 (Grebien et al, 2015) in two MCC cell lines (PeTa, MKL-1) and in HDFBs. The dose–response curves showed that BRD4 and TAF1 inhibitors are effective in the nM range (similar to the LSD1i), whereas WDR5 and PRMT5 inhibitors are only effective at μM concentrations (Fig EV2A–D). However, only the LSD1i selectively inhibited growth in MCC cell lines, while the four other tested drugs also impaired growth of HDFBs, indicating low specificity and general toxicity (Fig EV2A–D). Finally, we combined LSD1i (GSK-LSD1) with BRD4, PRMT5, TAF1, and WDR5 inhibitors, but found no significant synergistic effect at the tested combinations (Fig EV2A–D).

To assess the specificity of the LSD1 dependency, we compared MCC to other cancer types in the DepMap project (Tsherniak et al, 2017). While LSD1 is ubiquitously expressed in all tissues and cancer types (Fig EV2E and F), the expression level of LSD1 does not correlate with LSD1 dependency (Fig EV2G). We found that the median LSD1 dependency score of MCC was similar to that of hematopoietic and lymphoid malignancies (Fig 1F). Intriguingly, subtypes of hematopoietic and lymphoid cancers are known to respond to LSD1i in the clinic (Harris et al, 2012; Schenk et al, 2012). Collectively, our data show that genetic and pharmacological inhibition of LSD1 reduces MCC cell growth.

## Pharmacological LSD1 inhibition controls tumor growth *in vivo*

To assess the therapeutic efficacy of LSD1i *in vivo*, we subcutaneously injected PeTa cells into the flanks of immunocompromised NOD scid gamma mice (NSG). After the tumors reached a volume $\geq 50$ mm$^3$ (22 days post-injection), we treated the mice with GSK-LSD1 or vehicle (Fig 2A). We found that tumor growth was substantially reduced in mice treated with LSD1i compared to vehicle (Figs 2B and EV3A). All LSD1i-treated mice survived, whereas all vehicle-treated mice had to be euthanized due to high tumor burden (Fig 2C and D). Mouse weight remained stable throughout the treatment, suggesting that the mice tolerated the drug dose (Fig EV3B).

MCC is a highly metastatic cancer and may spread to several organs, contributing to its high morbidity and mortality (Kouzmina et al, 2017). Once seeded, micrometastases must grow and establish a tumor. To model the response of micrometastases, we subcutaneously injected MCC cells and started LSD1i treatment 1 day post-injection, prior to tumor establishment (Fig 2E). Whereas all (8/8) vehicle-treated mice grew tumors, all (8/8) mice treated with GSK-LSD1 (Figs 2F–H and EV3C) and seven of eight mice treated with ORY-1001 (Fig EV3D–H) remained tumor-free. Collectively, these results suggest that LSD1 inhibition reduces MCC growth *in vitro* and *in vivo*.

## LSD1 inhibition induces cell cycle arrest and cell death

To investigate the effects of LSD1i on MCC proliferation, we stained MCC cells with the proliferation marker Ki-67 and found ~ 3-fold downregulation after 6 days of LSD1i treatment *in vitro* (Fig 3A). Next, we assessed whether LSD1i treatment impairs cell cycle progression and performed EdU/PI labeling after 3 and 6 days of LSD1i *in vitro*. We found that after 3 days, in LSD1i-treated samples, cells in the G0/G1 phase population increased by ~ 7% compared to DMSO, while S and G2/M phase populations decreased each by ~ 25% compared to DMSO (Appendix Fig S1A). After 6 days of LSD1i treatment, the S phase population was decreased by 50%, while the initially increased G0/1 population was diminished, suggesting that LSD1i induces a G0/1 cell cycle arrest with subsequent cell death (Fig 3B and C). Interestingly, we detected no evidence for caspase-mediated cell death *in vitro*, indicated by a lack of cleaved PARP1 and cleaved caspase-3/7 (Fig 3D and E). To investigate other forms of cell death, we first assessed the loss of mitochondrial transmembrane potential ($\Psi$) as an early cell death marker, using tetramethylrhodamine ethyl ester (TMRE) (Crowley et al, 2016). We found that TMRE fluorescence decreased in a time-

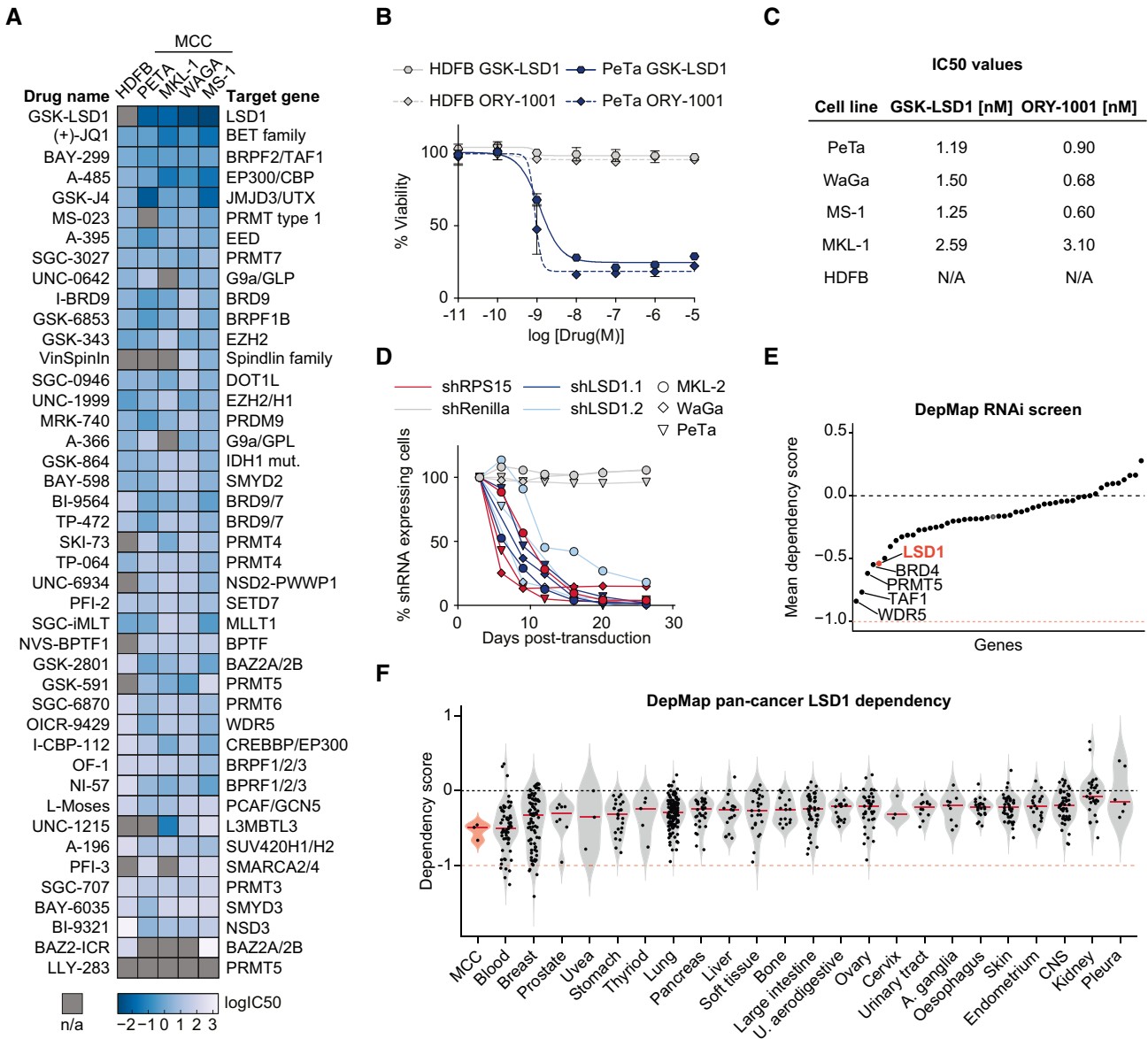

**Figure 1. LSD1 is required for Merkel cell carcinoma proliferation.**

A  Heatmap of IC50 values for cell viability. Human dermal fibroblasts (HDFB) controls and four MCC cell lines (PeTa, MKL-1, WaGa, and MS-1) were treated with the 43 indicated small molecules targeting epigenetic modifiers. IC50 values are depicted as $\log_{10}$(IC50) in (mM). n/a, IC50 values could not be calculated. $n = 4$ technical replicates.

B  Dose–response curves of PeTa cells and control HDFB cells after 6 days of treatment with GSK-LSD1 or ORY-1001. Dose–response curves of three other MCC cell lines are displayed in Fig EV1A. $n = 4$ technical replicates. Data are represented as means $\pm$ SD.

C  Calculated IC50 values for reduced growth of PeTa, WaGa, MS-1, MKL-1, and HDFB controls based on Figs 1B and EV1A.

D  *In vitro* competition assay of the three MCC cell lines MKL-2, PeTa, and WaGa transduced with either shLSD1.1, shLSD1.2, shRenilla (negative control), or shRPS15 (positive control). Individual graphs are displayed in Fig EV1E.

E  Dependency plot depicting the mean dependency of the three MCC cell lines PeTa, MKL-1, and MKL-2 of the genes targeted by the compound library in Fig 1A. A score of 0 indicates that a gene is not essential; correspondingly −1 is comparable to the median of all pan-essential genes. Data obtained from DepMap; dependencies for the individual cell lines are displayed in Fig EV1F.

F  Violin plot depicting the LSD1 dependency score in MCC compared to cancer types from 23 tissues, ordered according to mean dependency score. Red horizontal line depicts the median. Data obtained from DepMap RNAi screen. Blood, hematopoietic and lymphoid tissue; U. aerodigestive, upper aerodigestive tract; A. ganglia, autonomic ganglia; CNS, central nervous system.

Source data are available online for this figure.

dependent manner over 6 days of LSD1i (Fig 3F and G). Annexin V staining revealed a ~ 5- and 10-fold increase in early and late apoptotic cells, respectively (Fig 3H and I). In TUNEL staining, a marker for DNA double-strand breaks and hallmark of cell death, we observed a ~ 2-fold increase in LSD1i- compared to DMSO-treated cells (Fig 3J and Appendix Fig S1B).

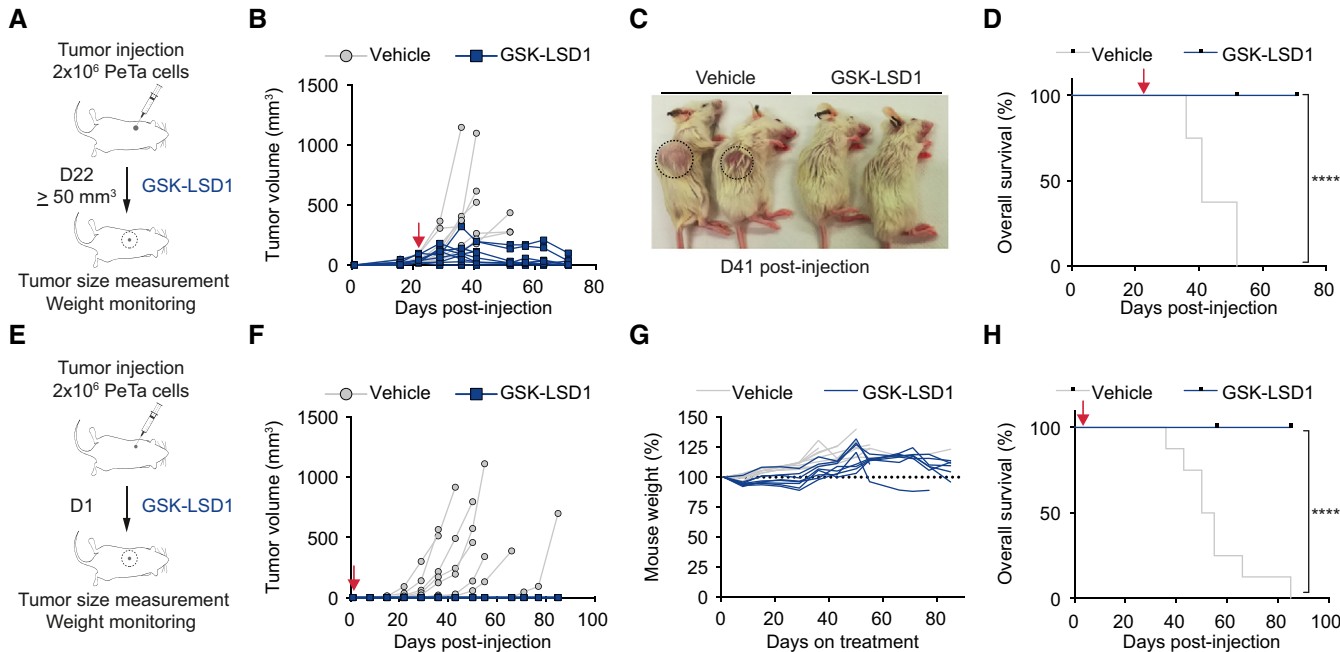

**Figure 2. Pharmacological LSD1 inhibition controls tumor growth *in vivo*.**

A   Schematic depicting the experimental setup for *in vivo* xenograft treatment of MCC tumors with GSK-LSD1 in NSG mice. GSK-LSD1 or vehicle treatment was started 22 days after PeTa cell injection, when tumor volume was ≥ 50 mm³.

B   Individual tumor growth in GSK-LSD1 (*n* = 9) or vehicle-treated (*n* = 8) mice. Red arrow: start of therapy, day 22.

C   Representative picture of mice treated with vehicle or GSK-LSD1 at day 41 after PeTa cell injection. Tumor location is indicated with a circle.

D   Kaplan–Meier curve of GSK-LSD1 (*n* = 9) or vehicle-treated (*n* = 8) mice. Mice were sacrificed when tumors reached a volume ≥ 1.5 cm³ or the greatest dimension was ≥ 1.5 cm. Red arrow: start of therapy, day 22. ****P < 0.0001 (log-rank Mantel–Cox test).

E   Schematic depicting the experimental setup for *in vivo* xenograft treatment of MCC "micrometastases" with GSK-LSD1 in NSG mice. GSK-LSD1 or vehicle treatment was started 1 day after tumor injection (D1).

F   Individual tumor growth in GSK-LSD1 (*n* = 8) or vehicle-treated (*n* = 8) mice. Red arrow: start of therapy, day 1.

G   Relative mouse weight (%) during treatment of GSK-LSD1 (*n* = 8) or vehicle-treated (*n* = 8) mice.

H   Kaplan–Meier curve of GSK-LSD1 (*n* = 8) or vehicle-treated (*n* = 8) mice. Mice were sacrificed when tumors reached a volume ≥ 1.5 cm³ or greatest dimension ≥ 1.5 cm. Red arrow: start of therapy, day 1. ****P < 0.0001 (log-rank Mantel–Cox test).

Source data are available online for this figure.

To confirm that these LSD1i-mediated effects on cell death and cell cycle contribute to reduced tumor growth *in vivo*, we performed Ki-67, TUNEL, and cleaved caspase-3 immunofluorescence (IF) stainings on tumors harvested after 1 day (D1), 10 days (D10), and at the experimental endpoint. At all three time points, we observed a ~ 50% decrease in cell proliferation (Ki-67, Fig 3K) and a ~ 5- to 10-fold increase in cell death upon LSD1 treatment (TUNEL, Fig 3L). In contrast to our observations *in vitro*, we also found a ~ 4-fold increased cleaved caspase-3 staining upon LSD1i treatment *in vivo* (Appendix Fig S1C). The H&E staining confirmed a decrease of mitotic cells and an increase in apoptotic bodies in the LSD1i-treated compared to vehicle-treated tumors (Fig 3M). Altogether, these data indicate that LSD1i treatment induces cell cycle arrest and cell death in MCC.

## LSD1 inhibition induces marked transcriptional changes in MCC

When investigating the effects of LSD1 on MCC growth, we noticed that MCC cells *in vivo* changed from relatively uniform, small, round to oval cells with round nuclei and scant cytoplasm in vehicle-treated mice to slightly larger and elongated cells with irregular-

shaped nuclei and ill-defined cell borders in LSD1-treated tumors (Fig 3M). *In vitro*, MCC cells became smaller and formed dense clusters upon LSD1i treatment, as opposed to their typical growth as loose aggregates in suspension (Fig 4A).

To dissect these morphological changes on a molecular level, we investigated the transcriptional changes and integrated them with the chromatin landscape. After treating PeTa cells for 6 days with LSD1i or DMSO, we performed RNAseq and found that LSD1i treatment led to the upregulation of 870 genes and the downregulation of 533 genes (Fig EV4A). In our CUT&RUN assays, we observed in the upregulated genes an increase of H3K4me1 and H3K27ac and a decrease of H3K27me3 at the proximal promoter, while global profiling of open chromatin (ATACseq) showed no difference of chromatin accessibility at the transcriptional start site. For downregulated genes, we observed an increase of H3K27me3 at the transcriptional start site coinciding with a slightly reduced chromatin accessibility after 6 days of LSD1i (Fig 4B and C). On protein level, we detected no global changes in histone modifications (Fig EV4B).

Pathway enrichment analysis of the downregulated genes was dominated by terms associated with cell cycle and DNA replication

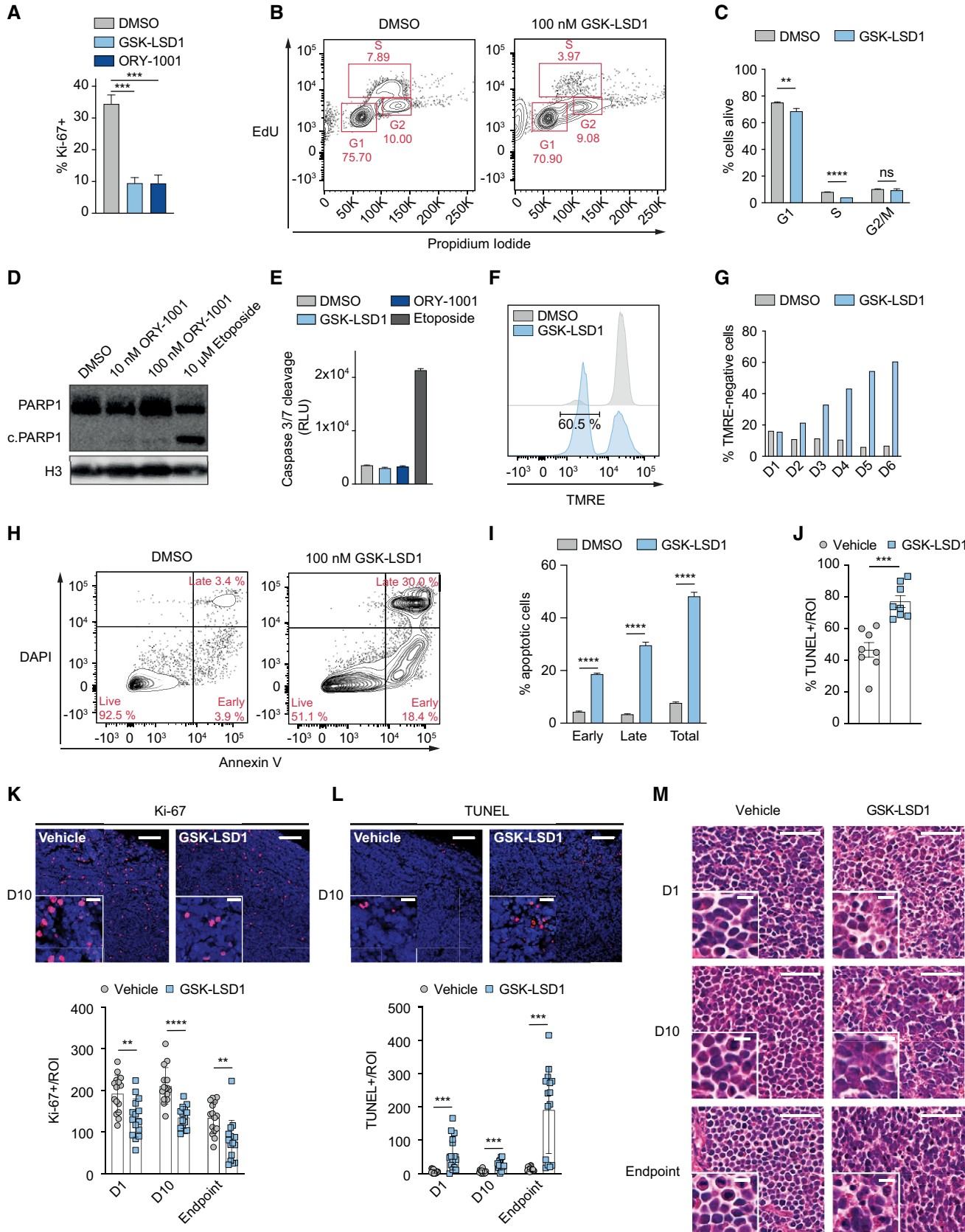

**Figure 3.**

**Figure 3.  LSD1 inhibition induces cell cycle arrest and cell death.**

A    Ki-67 staining of PeTa cells treated for 3 days with GSK-LSD1 (1 μM) or ORY-1001 (1 μM). $n = 3$ biological replicates. Data are represented as means ± SD. ***$P < 0.001$ (DMSO vs GSK-LSD1 $P = 0.0002$; DMSO vs ORY-1001 $P = 0.0004$; unpaired Student's $t$-test).

B    Representative FACS readout of EdU/propidium iodide cell cycle staining of PeTa cells after 6 days of 100 nM GSK-LSD1 or DMSO treatment.

C    Quantification of EdU/propidium iodide staining depicted in Fig 3B. $n = 3$ biological replicates. Data are represented as means ± SD. **$P < 0.01$; ****$P < 0.0001$; ns, non-significant (G1 phase: DMSO vs GSK-LSD1 $P = 0.0059$; S phase: DMSO vs GSK-LSD1 $P < 0.0001$; G2/M phase: DMSO vs GSK-LSD1 $P = 0.2338$; unpaired Student's $t$-test).

D    Immunoblot of PARP1 cleavage of PeTa cells after 6 days of indicated treatment. Etoposide serves as positive control for apoptosis, and H3 serves as loading control. c.PARP1, cleaved PARP1.

E    Caspase-3/7 cleavage activity for PeTa cell line after 6 days of indicated treatment. Etoposide serves as a positive control for apoptosis. $n = 3$ technical replicates. Data are represented as means ± SD. RLU, relative luminescence units.

F    Representative FACS readout of tetramethylrhodamine ethyl ester (TMRE) stained cells upon 100 nM GSK-LSD1 or DMSO treatment after 6 days.

G    Quantification of tetramethylrhodamine ethyl ester (TMRE) stained cells upon 100 nM GSK-LSD1 or DMSO treatment at indicated time points.

H    Representative FACS readout of Annexin V/DAPI staining of PeTa cells after 6 days of 100 nM GSK-LSD1 DMSO treatment.

I    Quantification of Annexin V/DAPI staining depicted in Fig 3H. $n = 3$ biological replicates. Data are represented as means ± SD. ****$P < 0.0001$ (Early, Late, Total; unpaired Student's $t$-test).

J    Quantification of *in vitro* TUNEL signal of PeTa cells after 6 days of 100 nM GSK-LSD1 or vehicle treatment. Representative images in Appendix Fig S1B. $n = 8$, ***$P = 0.001$ (Vehicle vs GSK-LSD1; unpaired Student's $t$-test). ROI, region of interest.

K    Top. Representative images of immunofluorescent Ki-67 staining of tumor slides after 10 days of *in vivo* GSK-LSD1 or vehicle treatment. Upper right scale bar represents 100 μm, and insert scale bar represents 20 μm. Bottom. Quantification of Ki-67 signal of tumor slides from mice treated with GSK-LSD1 or DMSO for 1 day, 10 days or until experiment endpoint. $n = 15$, **$P < 0.01$, ****$P < 0.0001$ (D1 vehicle vs D1 GSK-LSD1 $P = 0.0015$, unpaired Student's $t$-test; D10 vehicle vs D10 GSK-LSD1 $P < 0.0001$, unpaired Student's $t$-test; Endpoint vehicle vs Endpoint GSK-LSD1 $P = 0.0012$, Mann–Whitney test). ROI, region of interest.

L    Top. Representative images of immunofluorescent TUNEL staining of tumor slides after 10 days of *in vivo* GSK-LSD1 or vehicle treatment. Upper right scale bar represents 100 μm, and insert scale bar represents 20 μm. Bottom. Quantification of TUNEL signal of tumor slides from mice treated with GSK-LSD1 or DMSO for 1 day, 10 days or until experiment endpoint. $n = 15$, ***$P < 0.001$ (D1 vehicle vs D1 GSK-LSD1 $P = 0.0009$; D10 vehicle vs D10 GSK-LSD1 $P = 0.0001$; Endpoint vehicle vs Endpoint GSK-LSD1 $P = 0.001$; unpaired Student's $t$-test with Welch's correction). ROI, region of interest.

M    Representative images of hematoxylin and eosin (H&E) staining of tumor slides from mice treated with GSK-LSD1 or DMSO for 1 day (D1, top), 10 days (D10, middle) or until experiment endpoint (bottom). Upper right scale bar represents 50 μm, and insert scale bar represents 10 μm.

(Figs 4D and EV4C). We examined the motifs in the promoter region of downregulated genes and found motifs known to be involved in the regulation of cell cycle progression, such as the CHR (Müller *et al*, 2012), NFY (Benatti *et al*, 2011), and E2F (Mudryj *et al*, 1991) motifs enriching (Fig 4E), supporting our phenotypic observations on cell cycle arrest (Fig 3A–J). Pathway enrichment analysis of the upregulated genes was dominated by terms associated with nervous system development together with enrichment for the REST motif, which plays a role in repression of neuronal development and differentiation (Lunyak *et al*, 2002; Ballas *et al*, 2005) (Figs 4F and G, and EV4D).

**LSD1 directly represses transcription of key regulators of the neuronal lineage**

Intrigued by the finding that nervous system development dominated the pathway enrichment analysis of the upregulated genes, we asked whether LSD1 is a direct repressor of neuronal programs in MCC. To interrogate the direct transcriptional responses to LSD1i treatment, we treated MCC cells with vehicle or GSK-LSD1 for 30 min and performed SLAMseq, a metabolic RNA labeling method for time-resolved measurement of newly transcribed (nascent) RNA over 1 and 6 h (Herzog *et al*, 2017; Muhar *et al*, 2018) (Figs 5A and EV4E). We found a set of 22 genes with significantly increased transcription after just 1 h, which likely represent direct targets of LSD1 in MCC cells (Fig 5B). Interestingly, both the 1- and 6-h time points showed increased transcription of genes associated with neuron development (GO:0048666) and neurogenesis (GO:0022008), including the transcription factors NEUROD1 and INSM1, which regulate neuronal and neuroendocrine differentiation (Fig 5C) (Breslin *et al*, 2002; Pataskar *et al*, 2016). In line with the role of LSD1 as a transcriptional repressor, immediately upregulated LSD1 target genes

remained upregulated throughout LSD1i treatment, whereas the initial downregulation of the majority of LSD1 target genes was not maintained (Fig 5D).

Pathway enrichment analysis of the immediate transcriptional responses to LSD1 inhibition highlighted developmental and differentiation processes, as well as prominent clusters for TGFβ pathway members (Fig 5E). The TGFβ cluster included BMP receptors (NEO1, RGMA, RGMB), the downstream regulators SMAD9, and the ID family proteins ID1–4, all of which were robustly upregulated after 24 h of LSD1i treatment (Fig EV4F and G). Bone morphogenetic protein (BMP) signaling, as part of the TGFβ signaling pathway, regulates cell fate determination and has been associated with neuronal differentiation and innervation of Merkel cells (Meyers & Kessler, 2017; Jenkins *et al*, 2019).

Merkel cells are discussed to be a putative cell of origin of MCC (Harms *et al*, 2018). To test whether the transcriptional program of LSD1i-treated MCC cells resembles more closely that of differentiated and post-mitotic normal Merkel cells, we performed low RNA-input transcriptome analysis (SMARTseq) to analyze the transcriptomes of Merkel cells purified from mouse skin and of bulk skin cells (Fig 5F). Compared to the bulk skin transcriptome, the Merkel cell transcriptome was enriched for terms associated with neuronal differentiation and included genes important for normal Merkel cell development and maintenance, such as ATOH1, SOX2, and INSM1 (Fig 5G) (Perdigoto *et al*, 2014; Ostrowski *et al*, 2015; Rush *et al*, 2018). Additionally, we found NEUROD1, another transcription factor of the neuronal lineage that was previously only reported to be expressed in MCC, in the transcriptome of normal Merkel cells (Chteinberg *et al*, 2018; Rush *et al*, 2018). We defined a "Merkel cell signature" comprising the top differentially expressed genes and applied it to the transcriptome of vehicle- and LSD1i-treated MCC (Fig 5F). We found that the Merkel cell signature was highly

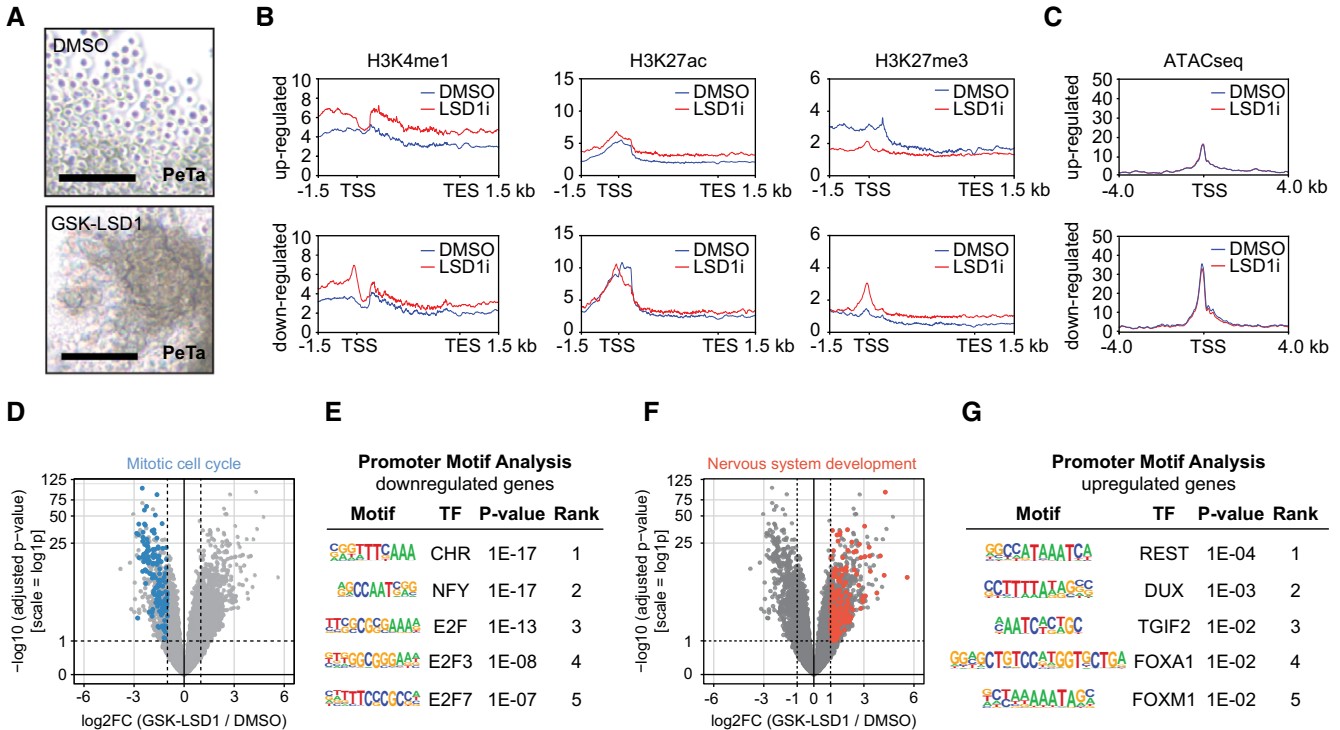

**Figure 4. LSD1 inhibition induces marked transcriptional changes.**

A Photomicrographs of the MCC cell line PeTa after 6 days of 100 nM GSK-LSD1 or DMSO treatment. Scale bar, 100 μm.
B Metagene plots showing the average CUT&RUN signal for H3K4me1, H3K27ac, and H3K27me3 for differentially expressed genes after 6 days of LSD1i or DMSO treatment. kb, kilobase; TSS, transcription start site; TES transcription end site.
C ATACseq peak profile around TSS of differentially expressed genes after 6 days of LSD1i or DMSO treatment. kb, kilobase; TSS, transcription start site.
D Volcano plot showing the $-\log_{10}$ (adjusted *P*-value) and $\log_2$ fold change ($\log_2$FC) for transcripts detected by RNAseq analysis of PeTa cells treated with 100 nM GSK-LSD1 or DMSO for 6 days. Significantly downregulated genes (FDR ≤ 0.05; $\log_2$FC ≤ −1) involved in mitotic cell cycle (GO:0000278) are highlighted in blue.
E Upstream promoter motif enrichment of the downregulated genes from the dataset in Fig 4D. Top-5 enriched motifs are depicted. TF, transcription factor.
F Volcano plot showing the $-\log_{10}$ (adjusted *P*-value) and $\log_2$ fold change ($\log_2$FC) for transcripts detected by RNAseq analysis of PeTa cells treated with 100 nM GSK-LSD1 or DMSO after 6 days. Significantly upregulated genes (FDR ≤ 0.05; $\log_2$FC ≥ 1) involved in nervous system development (GO:0007399) are highlighted in red.
G Upstream promoter motif enrichment of the upregulated genes from the dataset in Fig 4F. Top-5 enriched motifs are depicted. TF, transcription factor.

Source data are available online for this figure.

enriched in LSD1i-treated MCC cells (Fig 5H), suggesting that LSD1i treatment of MCC induces a transcriptional program recapitulating that of normal Merkel cells.

## HMG20B is an essential LSD1-CoREST complex subunit necessary for proliferation

LSD1 acts in cell context-specific protein complexes to regulate gene expression by demethylating H3K4me1/2, H3K9me1/2, and non-histone proteins (Andrés *et al*, 1999; Wang *et al*, 2007; Adamo *et al*, 2011). To uncover LSD1 binding partners in MCC, we immunoprecipitated LSD1, identified interacting proteins by mass spectrometry, and performed unsupervised protein–protein interaction enrichment analysis (Fig 6A–C). We found that in MCC, LSD1 interacts with members of the LSD1-CoREST complex (also called BHC, BRAF–histone deacetylase complex). In particular, we identified core members, including the histone deacetylase HDAC2, and RCOR1, RCOR2, and RCOR3, which serve as a scaffold for complex assembly, as well as non-canonical members, including GSE1, HMG20A, and HMG20B (Figs 6B and C, and EV5A).

Importantly, we found that LSD1i treatment of MCC leads to dissociation of this complex, most notably loss of HMG20B binding (Figs 6D and EV5B). While pharmacological disruption of the LSD1-CoREST complex is known to displace complex subunits, such as SNAG domain-containing proteins (Saleque *et al*, 2007; Ferrari-Amorotti *et al*, 2013; Maiques-Diaz *et al*, 2018; Egolf *et al*, 2019), the loss of the HMG20B protein subunit has not been reported before.

To determine whether the LSD1 binding partners are also required for cell growth in MCC, we examined the RNAi data from the DepMap project (Tsherniak *et al*, 2017). Our analysis suggested that LSD1, the scaffolding protein RCOR1, and HMG20B are specifically required for MCC growth compared to other skin cancers (Figs 6E and EV5C–E). We performed quantitative mass spectrometry (TMT-MS) on cell nuclei treated for 24 h with LSD1i or DMSO to investigate whether the disruption of the nuclear LSD1-CoREST complex by LSD1i also changes their protein abundance (Fig 6F). We identified 26 significantly upregulated and three significantly downregulated proteins, of which HMG20B was the most significantly downregulated (50%) (Fig 6G and H).

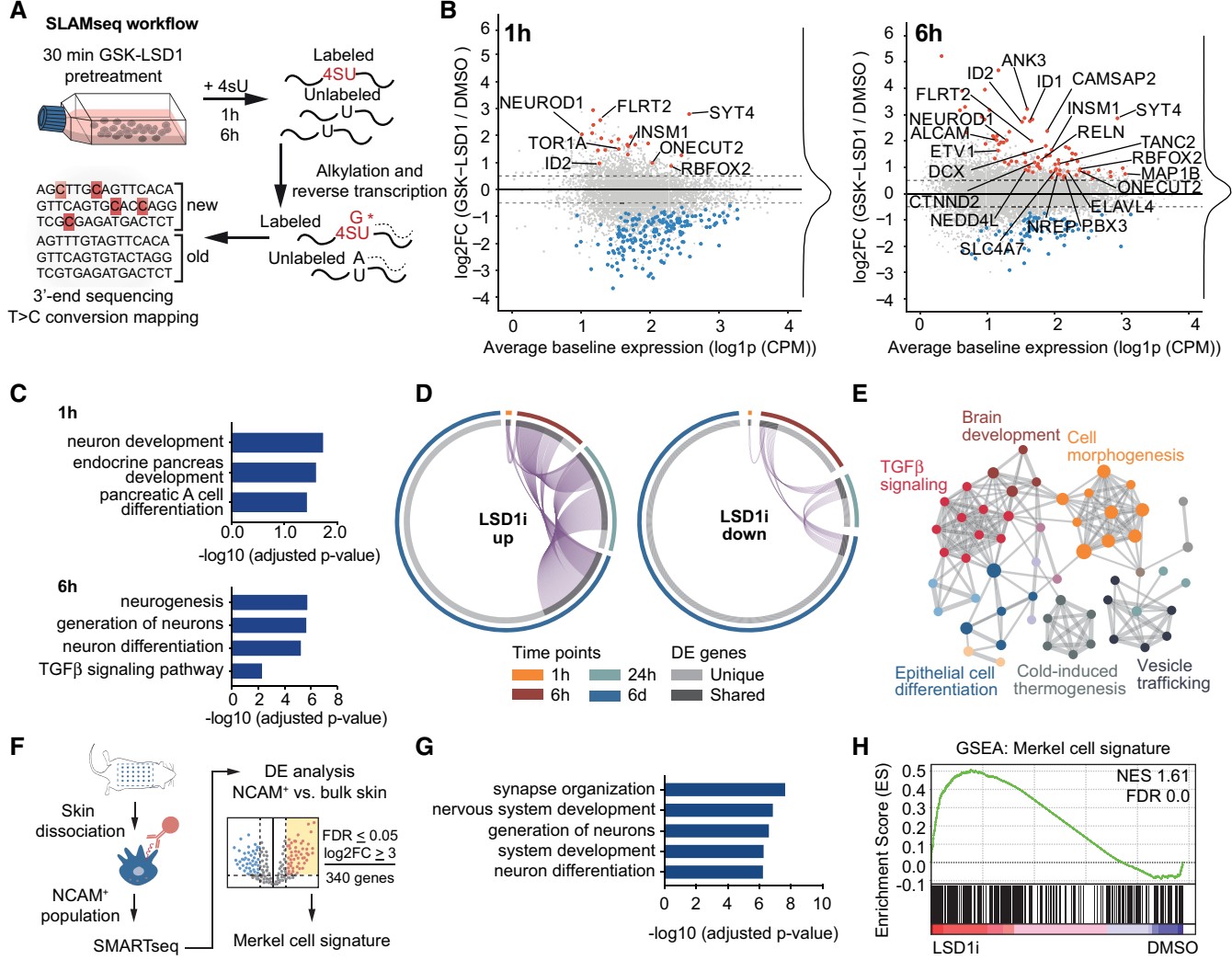

**Figure 5. LSD1 directly represses transcription of key regulators of the neuronal lineage.**

A  Schematic of SLAMseq workflow. Cells were pre-treated for 30 min with LSD1 inhibitor (100 nM GSK-LSD1), and newly synthesized RNA was subsequently labeled for 1 or 6 h with 4-thiouridine (4sU). RNA was extracted and alkylated and subjected to 3′-end RNA sequencing.

B  MA plot of the mRNA changes detected by SLAMseq after 1 and 6 h of 4sU labeling. Significantly up- and downregulated genes (FDR ≤ 0.05; abs [log$_2$FC] ≥ 0.5) are marked in red and blue, respectively. Genes belonging to neuronal differentiation (GO:0048666) are labeled. FC, fold change; CMP, counts per million.

C  Pathway enrichment analysis for upregulated (FDR ≤ 0.05; log$_2$FC ≥ 0.5) direct transcriptional targets of LSD1 in Fig 5B.

D  Circos plots depicting the transcriptional changes of genes up- (up) and downregulated (down) upon LSD1i after 1 h, 6 h, 24 h, and 6 days. DE, differentially expressed.

E  Metascape analysis of upregulated genes for enriched pathway terms in genes identified after 6 h of 4sU-labeling. Major pathway clusters are labeled.

F  Schematic of Merkel cell extraction and Merkel cell signature generation. DE, differential expression; FDR, false discovery rate; log$_2$FC, log$_2$ fold change.

G  Pathway enrichment analysis (GO:BP) of genes upregulated in Merkel cell signature.

H  Gene set enrichment analysis (GSEA) of the generated Merkel cell signature on the data in Fig 4B. NES, normalized enrichment score; FDR, false discovery rate.

Source data are available online for this figure.

HMG20B is ubiquitously expressed and required to maintain full repression of neuron-specific genes as part of the CoREST complex (Hakimi *et al*, 2002; Ceballos-Chávez *et al*, 2012). Notably, about half of the upregulated proteins were associated with nervous system development (GO:0007399), including SYT4, ANK3, and SOX3, identified by SLAMseq (Fig 5B). We also identified an upregulation of NCAM1 and GAP43 proteins, both of which are expressed in normal Merkel cells (Gallego *et al*, 1995; Verzé *et al*, 2003). Together with the downregulation of HMG20B on the protein level

(Fig 6H) and in the absence of a transcriptional deregulation (Fig EV5F and G), we conclude that pharmacological inhibition of LSD1 affects LSD1-CoREST complex assembly and protein stability.

To understand the role of HMG20B in MCC, we performed a protein domain analysis of the human HMG20B protein which is distinct from the mammalian HMG box domain-containing protein family (Sumoy *et al*, 2000). We found that HMG20B is characterized by an HMG (high-mobility group) box domain (aa 70–137, (El-Gebali *et al*, 2019)), a coiled-coil region (aa 196–251, (Lupas *et al*,

1991)) two highly conserved alpha-helices (aa 276–312), and a predicted nuclear localization signal (aa 55–65, (Kosugi *et al*, 2009)) (Fig 6I). To delineate the contributions of the different HMG20B

domains to LSD1-CoREST complex integrity and cell survival in MCC, we overexpressed V5-tagged domain deletion mutants of HMG20B (Fig 6J). We performed a co-immunoprecipitation

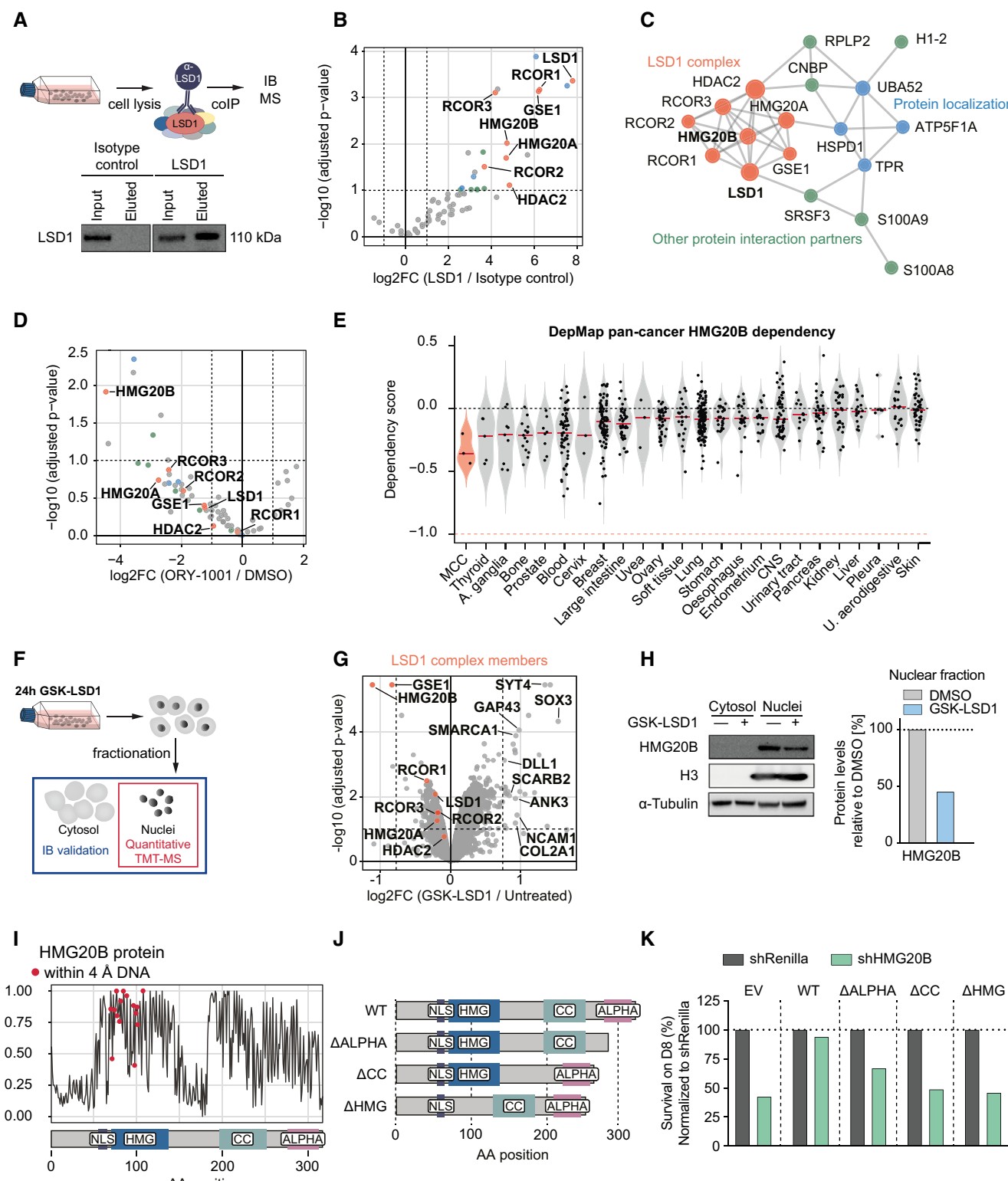

**Figure 6.**

**Figure 6. HMG20B is an essential LSD1-CoREST complex subunit necessary for proliferation.**

A  Schematics and immunoblot of LSD1 co-immunoprecipitation (co-IP) from PeTa cells. IB, Immunoblot; MS, mass spectrometry.
B  Volcano plot displaying the identified protein–protein binding partners of LSD1. Proteins belonging to the LSD1 complex are marked in red, protein localization in blue, and other protein interaction partners in green. FC, fold change.
C  Protein–protein interaction (PPI) mapping of the identified LSD1 binding partners. Individual complexes and *P*-values are displayed in Fig EV5A.
D  Volcano plot depicting the protein–protein binding partners of LSD1 depleting in PeTa cells treated with ORY-1001 vs DMSO. Proteins belonging to the LSD1 complex are marked in red, protein localization in blue, and other protein interaction partners in green. FC, fold change.
E  Violin plot depicting the HMG20B dependency in MCC compared to cancer types from 23 tissues, ordered according to mean. Red horizontal line depicts the median. Data obtained from DepMap RNAi dataset. Blood, hematopoietic and lymphoid tissue; U. aerodigestive, upper aerodigestive tract; A. ganglia, autonomic ganglia; CNS, central nervous system.
F  Schematic of quantitative TMT-MS experiment. PeTa cells were treated for 24 h with 100 nM GSK-LSD1 or DMSO. Cells were fractionated to separate nuclei from cytosol. Nuclei were analyzed by TMT-MS. IB, Immunoblot; TMT-MS, tandem mass tag labeling mass spectrometry.
G  Volcano plot depicting the differentially regulated proteins obtained from Fig 6F. Proteins members of the LSD1-CoREST complex are labeled and highlighted in red. Proteins significantly upregulated ($-\log_{10}[P\text{-adjusted}] \geq 1$; $\log_2 FC \geq 0.75$) and involved in nervous system development (GO:0007399) are labeled. FC, fold change.
H  Left. Immunoblot confirmation of HMG20B downregulation. H3 serves as nuclear fraction control and α-Tubulin as loading control. Right. Immunoblot quantification. HMG20B protein levels are normalized to H3 and relative to DMSO control.
I  Amino acid conservation analysis of HMG20B across species and (bottom) the corresponding domains. Predicted DNA interacting amino acids are indicated with a red dot. aa, amino acid; NLS, nuclear localization signal (aa: 55–65); HMG, high-mobility group box (aa: 70–137); cc, coiled-coil domain (aa: 196–251); ALPHA, alpha-helices (aa: 276–312).
J  Schematic of HMG20B domain deletion mutant proteins. WT, wild-type; ΔALPHA, alpha-helices deletion; ΔCC, coiled-coil domain deletion; ΔHMG, HMG box deletion.
K  Bar graph of HMG20B rescue experiment in PeTa cells transduced with either a shRNA targeting HMG20B (shHMG20B) or negative control (shRenilla) and with an overexpression construct expressing either of the HMG20B mutants that cannot be targeted by the shRNA. Cell survival is depicted at day 8 after transduction and normalized to shRenilla. EV, empty vector control, WT, wild-type; ΔALPHA, alpha-helices deletion; ΔCC, coiled-coil domain deletion; ΔHMG, HMG box deletion.

Source data are available online for this figure.

experiment of the different mutants which revealed that the coiled-coil region of HMG20B is essential for binding to the LSD1-CoREST complex (Fig EV5H). In contrast, neither the deletion of the DNA binding HMG box domain nor of the two alpha-helices changed binding to the LSD1-CoREST complex. To verify that HMG20B is required for MCC growth, we depleted endogenous HMG20B. Indeed, the depletion of HMG20B reduced MCC cell growth, which was rescued by expression of a wild-type HMG20B but not by any of the HMG20B domain deletion mutants that were not targeted by the shRNA (Fig 6K). Finally, we assessed whether combinatorial pharmacological targeting of multiple LSD1-CoREST complex members has a synergistic effect on treatment response by combining histone deacetylase (HDAC) and LSD1 inhibitors, as this has been shown for other cancers (Kalin *et al*, 2018; Anastas *et al*, 2019). However, we found no synergistic effect combining Entinostat (HDAC1/3) or Santacruzamate A (HDAC2) inhibitors with GSK-LSD1 (Fig EV5I and J). Together, these data indicate that an LSD1-RCOR1-HMG20B complex is required for MCC growth.

**Pharmacological LSD1i induces a durable shift in cell fate in MCC**

To investigate whether short-term LSD1i treatment leads to a durable shift in cell fate, we treated PeTa cells for 24 h with 100 nM GSK-LSD1 or DMSO, washed out the drug, and harvested cells for 8 days (Fig 7A). We found that a 24 h LSD1i-pulse was sufficient to induce sustained transcriptional upregulation of LSD1 targets of the neuronal lineage, such as ANK3, SOX3, SYT4, or NEUROD1 and derepression of the BMP-arm of TGFβ signaling, indicated by elevated levels of phospho-SMAD1/5/9 and ID1 over 8 days (Fig 7B–C). Additionally, we found that HMG20B protein levels were strongly decreasing over 8 days suggesting a maintained disruption of the LSD1-CoREST complex (Fig 7D). For other LSD1-CoREST complex members, we did not observe changes in protein abundance (Fig 7D). To uncover if maintained activation of those programs is sufficient to induce a reduction in cell proliferation, we

performed EdU/PI and Annexin V/DAPI stainings. Similar to our previous results obtained in the continuous presence of LSD1i (Fig 3B, C, H and I), we observed a G0/1 cell cycle arrest (Fig 7E and F) together with strong induction of cell death (Fig 7C and G). Finally, we probed whether the durable shift in cell fate affects tumor formation *in vivo*. We pre-treated PeTa cells, either for 1, 3, or 6 days with 100 nM GSK-LSD1 or DMSO, and found that pre-treated cells have a strongly reduced tumor formation propensity (Fig 7H and I). Altogether, these data indicate that short-term treatment with LSD1i induces a durable shift in cell fate in MCC, driven by the disassembly of the LSD1-CoREST complex and a sustained activation of master regulators of the neuronal lineage.

# Discussion

Therapies that target cell fate regulators, instead of aberrantly activated oncogenic drivers, such as kinases, have not been extensively explored in solid cancers, but demonstrate clinical activity in leukemias, e.g. all-trans retinoic acid in acute promyelocytic leukemia (Huang *et al*, 1989; Tallman *et al*, 1997). Here, we performed a pharmacological drug screen to identify epigenetic cell fate regulators in virus-positive MCC, which lacks druggable driver mutations. Our work reveals that the ubiquitously expressed histone demethylase LSD1 is a dependency and potential therapeutic target in MCC. In MCC, LSD1 directly represses the expression of master regulators of the neuronal lineage and members of the BMP/TGFβ signaling cascade. Pharmacological inhibition of LSD1 leads to an activation of a neuronal differentiation program, driving MCC cells toward a normal Merkel cell fate and inducing cell cycle arrest and cell death. We show that LSD1 inhibitors lead to the disruption of the LSD1-CoREST complex and the displacement and eventual degradation of the subunit HMG20B, similar to SNAG domain-containing proteins that get released from the LSD1-CoREST complex upon LSD1 inhibition in other cellular contexts (Ferrari-Amorotti *et al*, 2013;

Maiques-Diaz *et al*, 2018; Egolf *et al*, 2019). Notably, HMG20B has previously been implicated in the repression of neuronal lineage-specific genes (Hakimi *et al*, 2002; Ceballos-Chávez *et al*, 2012).

Recently, Park *et al* reported that the small T antigen, one of the oncogenic drivers of the Merkel cell polyomavirus, directly induces expression of LSD1-CoREST complex members LSD1 and RCOR2, as well as of the transcription factor INSM1, which we identify as a target of LSD1. Intriguingly, they found that LSD1 and RCOR2 binding sites overlap with those of ATOH1, a key transcription factor of normal Merkel cell development (Bardot *et al*, 2013; Park *et al*, 2020). Other studies found that the inhibition of the T antigens induces cell cycle arrest (Houben *et al*, 2010) and induces neuronal differentiation when co-cultured with keratinocytes (Harold *et al*, 2019). Collectively, these data suggest that T antigen-mediated transformation relies on LSD1 to suppress differentiation toward a post-mitotic Merkel cell fate and lock MCC cells in a stem-like state.

Putting our findings in MCC in the broader context of studies investigating LSD1i treatment in other cancers, such as AMLs (Schenk *et al*, 2012; Somervaille *et al*, 2016; Fang *et al*, 2019; Cai *et al*, 2020), SCLC (Mohammad *et al*, 2015; Augert *et al*, 2019), prostate cancer (Sehrawat *et al*, 2018), and cutaneous squamous cell carcinoma (Egolf *et al*, 2019), it appears that in certain cellular lineages, the neuroendocrine lineage among them, LSD1 is a gate-keeper of lineage plasticity. Inhibition of LSD1 may therefore represent a tractable entry point for differentiation therapies of solid cancers.

There is growing evidence that cancer cells exploit cellular plasticity and dedifferentiation programs to evade destruction by the immune system (Li & Stanger, 2020). Immunotherapy with checkpoint inhibitors, such as anti-PDL1/-PD1, is the first-line treatment for MCC but a large fraction of patients do not respond (Kaufman *et al*, 2016; Nghiem *et al*, 2016). It is tempting to speculate that differentiation of MCC using LSD1 inhibitors could enhance responsiveness to checkpoint inhibitors. While there are no immunocompetent MCC mouse models available to investigate whether LSD1i enhances an immune-response, this is in line with a recent report indicating that LSD1 depletion enhances response to checkpoint inhibitors by activating type 1 interferon signaling, which stimulates T-cell responses (Sheng *et al*, 2018). The combination of LSD1i with immunotherapies may thus represent a promising therapeutic avenue for MCC and other cancers.

## Materials and Methods

### DepMap and TCGA data analysis

The DepMap (https://depmap.org/portal/) RNAi screen dataset (rnai_19Q1, EH2260) was analyzed using the Bioconductor R package "depmap" (Killian & Gatto, 2019). Individual or mean dependencies for genes targeted in the epigenetic modifier screen were calculated for the three MCC cell lines (MKL-1, MKL-2, and PeTa). LSD1, HMG20B, RCOR1, and GSE1 dependency scores across all cancer cell lines were grouped by tissue type as pre-defined by DepMap. MCC cell lines were removed from the tissue type "SKIN" and grouped as "MCC". Tissue types with less than three

cell lines or cell lines with less than three dependency scores for a specific gene were removed from the dataset. For dependency expression plots, CCLE expression data (RSEM, gene; 02-Jan-2019) were correlated with the DepMap RNAi screen dataset (rnai_19Q1, EH2260). TCGA tumor and normal tissue RNAseq expression data were obtained from the Ordino database (version 6.0.1, (Streit *et al*, 2019)), and tissues with less than three samples were removed from the dataset. All plots were generated with the R package "ggplot2" (Wickham, 2016).

### Cell culture

The MCC cell lines, MKL-1, MKL-2, MS-1, PeTa, and WaGa, were cultured as suspension cells in RPMI-1640 (21875091, Thermo Fisher Scientific) supplemented with 10% FBS (F7524, Sigma-Aldrich), 2 mM L-glutamine (25030081, Thermo Fisher Scientific), 50 U/ml penicillin, and 50 mg/ml streptomycin (P0781, Sigma-Aldrich). HDFB and HEK-293T (Lenti-X) cells were cultured as an adherent monolayer in DMEM high glucose medium (in-house) supplemented with 10% FBS, 2 mM L-glutamine, 50 U/ml penicillin, and 50 mg/ml streptomycin. All cells were maintained at 37°C and 5% $CO_2$. All MCC cell lines were validated by STR profiling. Cells were kept at a low passage and tested for mycoplasma regularly. The MCC cell lines (MKL-1, MKL-2, MS-1, PeTa, WaGa) used in this study were a kind gift from Dr. David Schrama (University Wuerzburg, Germany). HEK-293T cells were purchased from Takara (Lenti-X 293T, 632180). Primary HDFB cells were purchased from ATCC (PCS-201-010).

### Epigenetic modifier screen

Suspension cells were seeded at a density of 10,000 cells per well; adherent cells were plated at a density of 2,500 cells per well in a 96-well plate. The epigenetic probes collection, obtained from the Structural Genomics Consortium (http://www.thesgc.org), was dissolved in DMSO, and compounds were probed at concentrations of 100 μM, 1 μM, 10 nM, and DMSO only. Cell viability was determined with the CellTiter-Glo® 2.0 Cell Viability Assay (G9242, Promega) according to the manufacturer's instructions on day 6 after seeding. Dose–response curves were generated in quadruplicates. IC50 values were calculated using the R package "GRmetrics" (Clark *et al*, 2017).

### Dose–response curves

Suspension cells were seeded at a density of 10,000 cells per well; adherent cells were plated at a density of 2,500 cells per well in a 96-well plate. ORY-1001 (S7795, Selleck Chemicals) and GSK-LSD1 (S77574, Selleck Chemicals) were resuspended in DMSO and serially diluted with final concentrations ranging from 0.01 nM to 100 μM. Cells were treated in quadruplicates at indicated doses or DMSO for 6 days. Cell viability was read out with the CellTiter-Glo® 2.0 Cell Viability Assay (G9242, Promega) according to the manufacturer's instructions. IC50 curves were calculated with the software GraphPad PRISM 8 (non-linear regression, log (inhibitor vs response—variable slope, four parameters)).

## Drug synergy maps

Suspension cells were seeded at a density of 10,000 cells per well; adherent cells were plated at a density of 2,500 cells per well in a 96-well plate. Cells were treated in quadruplicates with GSK-LSD1 at final concentrations of 1, 10 or 100 nM, or DMSO in combination with a second drug of final concentrations ranging from 1 nM to 10 µM or DMSO for 6 days. The second drug was either (+)-JQ1, OICR9429, GSK591, or BAY299 from the epigenetic probes collection, Entinostat MS-275 (S1053, Selleck Chemicals), or Santacruzamate A CAY10683 (S7595, Selleck Chemicals). Cell viability was read out with the CellTiter-Glo® 2.0 Cell Viability Assay (G9242, Promega) according to the manufacturer's instructions. IC50 curves were calculated with the software GraphPad PRISM 8 (non-linear

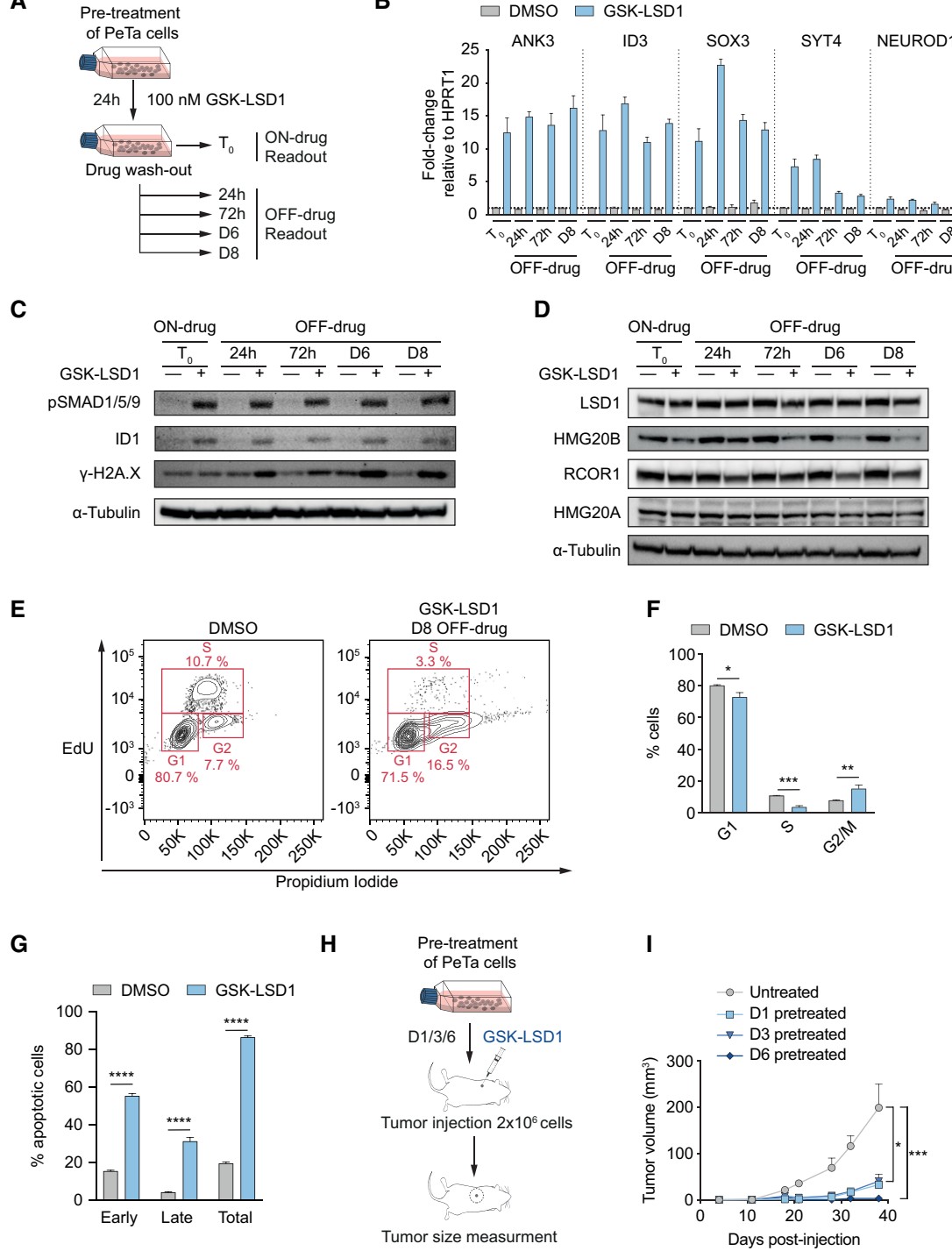

**Figure 7.**

◀

**Figure 7.  Pharmacological LSD1 inhibition induces a sustained change in cell fate.**

A   Schematic of drug pulse and wash-out experiment. PeTa cells were treated with 100 nM GSK-LSD1 or DMSO. After 24 h, a cell sample was harvested ($T_0$, ON-drug) and the remaining cells were washed and changed into fresh medium without drug. Additional cell samples were taken 24 h, 72 h, 6 days (D6), and 8 days (D8) after drug wash-out (OFF-drug) for downstream readouts.
B   RT–qPCR quantification of neuronal genes at the different time points and conditions depicted in Fig 7A. Data are relative to the housekeeping gene HPRT1 and normalized to the respective DMSO control. $n = 4$ technical replicates. Bar graphs represent mean $\pm$ SD.
C   Immunoblot probing for sustainability of LSD1 activity inhibition at the different time points and conditions depicted in Fig 7A. α-Tubulin serves as loading control. pSMAD1/5/9, phospho-SMAD1/5/9.
D   Immunoblot probing for LSD1-CoREST complex members at the different time points and conditions depicted in Fig 7A. α-Tubulin serves as loading control.
E   Representative FACS readout of EdU/propidium iodide cell cycle staining of PeTa cells 8 days after drug wash-out.
F   Quantification of EdU/propidium iodide staining depicted in Fig 7E. $n = 3$ biological replicates. Data are represented as means $\pm$ SD. *$P < 0.05$, **$P < 0.01$; ***$P < 0.001$ (G1 phase: DMSO vs GSK-LSD1 $P = 0.0104$; S phase: DMSO vs GSK-LSD1 $P = 0.0001$; G2/M phase: DMSO vs GSK-LSD1 $P = 0.0062$; unpaired Student's *t*-test).
G   Quantification of FACS Annexin V/DAPI staining of PeTa cells 8 days after drug wash-out. $n = 3$ biological replicates. Data are represented as means $\pm$ SD. ****$P < 0.0001$ (unpaired Student's *t*-test).
H   Schematic depicting the experimental setup of the *in vivo* tumor growth of pre-treated cells. PeTa cells were pre-treated *in vitro* with 100 nM GSK-LSD1 for 1, 3 or 6 days, or DMSO treated for 6 days prior injection.
I    Subcutaneous tumor growth of pre-treated cells (untreated, $n = 8$; D1, $n = 9$; D3, $n = 9$; D6, $n = 4$). Data are represented as means $\pm$ SEM. *$P < 0.05$, ***$P < 0.001$ (untreated vs D1 $P = 0.023$; untreated vs D3 $P = 0.023$; untreated vs D6 $P = 0.0007$; Kruskal–Wallis test).

Source data are available online for this figure.

regression, log [inhibitor vs response—variable slope, four parameters]). Drug synergy maps were calculated with the R Bioconductor package "synergyfinder" (He *et al*, 2018).

### Photomicrographs

Merkel cell carcinoma PeTa cells were seeded at a density of 10,000 cells per well in a 96-well plate. Cells were treated with GSK-LSD1 (100 nM) or DMSO. Photomicrographs were taken on day 6. Scale bar was defined with ImageJ.

### Lentivirus production and cell transduction

For lentivirus production, HEK-293T (Lenti-X) cells were co-transfected with the lentiviral backbone plasmid of interest, VSVG as enveloping plasmid, and the packaging plasmid PAX2 in standard medium with polyethyleneimine (PEI) as previously reported (Muhar *et al*, 2018). Twenty-four hours after transfection, the culture medium was changed to 1% FBS medium and viral supernatant was collected after 24 h and filtered through a 0.4-μm mesh. Transduction of cell lines was performed by spinfection at 800 *g* for 30 min at 32°C with 10 μg/ml polybrene.

### shRNA cloning and competition assay

shRNAs were designed with the *splashRNA* software (Pelossof *et al*, 2017) and cloned into the backbone by Gibson assembly as previously described (Fellmann *et al*, 2013). Cells ($3 \times 10^6$) were transduced at a multiplicity of infection (MOI) of ~ 1 with shRNAs cloned into a doxycycline-inducible, GFP-expressing, and puromycin selectable backbone (T3G-GFP-miRE-PGK-Puro-IRES-rtTA3, LT3GEPIR), resulting in ~ 30% GFP-positive cells at day 0 of the experiment. The shRNA expression was induced with 500 ng/ml doxycycline. The relative abundance of transduced cells (GFP-positive cell population vs non-transduced population) was followed over time by flow cytometry. Twenty thousand single-cell events were acquired per sample on a FACS LSR Fortessa cytometer, and GFP-positive fractions were determined with the FlowJo™10

software. shRNA hairpins targeting Renilla luciferase and RPS15 were used as non-targeting and killing control, targeting a core-essential gene, respectively.

The 22mer shRNA sequences are as follows:

- shRenilla, Renilla-luciferin 2-monooxygenase gene:
  5′-TAGATAAGCATTATAATTCCTA-3′
- shRPS15, RPL15 gene:
  5′-TATAACGTAACCTTGCTTGGCC-3′
- shLSD1.1, LSD1/KDM1A gene:
  5′-TTAAGAAGTTCTTTCAATTCTT-3′
- shLSD1.2, LSD1/KDM1A gene:
  5′-TAATTCATCATATTCCTTGCAT-3′
- shHMG20B, HMG20B/BRAF35 gene:
  5′-TTTCTTGATCTTCTTCTCCTGG-3′

### Immunoblotting

Cells were lysed with RIPA buffer (9806, CST) supplemented with cOmplete™ Protease Inhibitor Cocktail (5056489001, Sigma-Aldrich) and HALT™ phosphatase inhibitor (78427, Thermo Fisher Scientific). Lysates were sonicated and cleared by centrifugation at 14,000 *g* for 10 min at 4°C. Protein concentrations were determined with the Pierce™ BCA Protein Assay kit (23227, Thermo Fisher Scientific) according to the manufacturer's instructions. Immunoblotting was conducted according to standard protocols. Protein levels on immunoblots were quantified with ImageJ and normalized to the loading control. The primary antibodies used for immunoblotting were as follows: anti-Vinculin (V9131, Sigma-Aldrich, 1:1,000), anti-LSD1 (2139, CST, 1:2,000), anti-phospho-SMAD1/5/9 (138820, CST, 1:1,000), anti-ID1 (sc-133104, Santa Cruz Biotechnology, 1:250), anti-ID2 (3431s, CST, 1:250), anti-V5 (R960-25, 1:5,000), anti-PARP1 (9532, CST, 1:1,000), anti-H3 (ab1791, Abcam, 1:5,000), anti-H3K4me1 (C15410194, Diagenode, 1:1,000), anti-H3K4me2 (9725, CST, 1:2,000), anti-H3K27ac (ab4729, Abcam, 1:2,000), H3K27me3 (97733, CST, 1:2,000), anti-HMG20B (14582-1-AP, Proteintech, 1:500), anti-HMG20A (12085-2-AP, Proteintech, 1:500), anti-RCOR1/CoREST (07-455, Millipore, 1:1,000), anti-

GSE1/KIAA0182 (24947-1-AP, Proteintech, 1:500), anti-α-tubulin (2125S, CST, 1:1,000), and anti-γ-H2A.X phospho-S139 (ab2893, Abcam, 1:1,000). The secondary antibodies used were as follows: anti-rabbit IgG HRP-linked (7074S, CST, 1:5,000), anti-mouse IgG HRP-linked (7076S, CST, 1:5,000), and anti-mouse IgG light chain-specific HRP-linked (58802S, CST, 1:5,000).

## RNA extraction

Total RNA isolation from cells was performed with an in-house paramagnetic bead-based purification protocol. Briefly, cells were lysed, incubated with functionalized paramagnetic beads, and stringently washed to remove proteins. Beads were then treated with DNaseI, and purified total RNA was eluted in nuclease-free water. Concentration was determined with the spectrophotometer/fluorometer DeNovix DS-11 Fx and checked for RNA integrity on a Fragment Analyzer System (Agilent).

## RT–qPCR

First-strand synthesis was performed with 1 μg of extracted total RNA either with SuperScript™ III Reverse Transcriptase (18080093, Invitrogen) or with the LunaScript™ RT SuperMix Kit (E3010, New England BioLabs) according to the manufacturer's instructions. The qRT–PCR was performed either with GoTac® qPCR (A6001, Promega) or with Luna® Universal qPCR Master Mix (M3003, New England BioLabs) with 10 ng cDNA template on a Biorad CFX384 Real-Time Cycler. Data were normalized to the housekeeping gene HPRT1 and displayed as relative to the control, shRenilla, or DMSO.

Primer sequences for RT–qPCR analysis are as follows:

- LSD1 forward primer: 5′-CTCTTCTGGAACCTCTATAAAGC-3′
- LSD1 reverse primer: 5′-CATTTCCAGATGATCCTGCAGCAA-3′
- HPRT1 forward primer: 5′-TGACACTGGCAAAACAATGCA-3′
- HPRT1 reverse primer: 5′-GGTCCTTTTCACCAGCAAGCT-3′
- NEUROD1 forward primer: 5′-ATGACCAAATCGTACAGCGAG-3′
- NEUROD1 reverse primer: 5′-GTTCATGGCTTCGAGGTCGT-3′
- SYT4 forward primer: 5′-ATGGGATACCCTACACCCAAAT-3′
- SYT4 reverse primer: 5′-TCCCGAGAGAGGAATTAGAACTT-3′
- ANK3 forward primer: 5′-GAAGATGCAATGACCGGGGA-3′
- ANK3 reverse primer: 5′-CTAAAGCCCATGTAACCCTCTG-3′
- ID3 forward primer: 5′-TCCTTTTGTCGTTGGAGATGAC-3′
- ID3 reverse primer: 5′-GAGAGGCACTCAGCTTAGCC-3′
- SOX3 forward primer: 5′-GACCTGTTCGAGAGAACTCATCA-3′
- SOX3 reverse primer: 5′-CGGGAAGGGTAGGCTTATCAA-3′

## LSD1-immunoprecipitation (co-IP)

Cells were pre-treated with either 1 μM ORY-1001 or DMSO 24 h prior to the co-IP. Cells were washed with ice-cold PBS and lysed on ice with 1× Cell Lysis Buffer (9803, CST). Lysates were cleared by centrifugation at 14,000 g for 10 min at 4°C. The lysate concentration was determined with the Pierce™ BCA Protein Assay kit (23227, Thermo Fisher Scientific) according to the manufacturer's instructions. Protein A sepharose magnetic beads (GE28-9670-56, Sigma-Aldrich) were pre-washed with 1× Cell Lysis Buffer. To reduce non-specific binding to the beads, the lysates were pre-cleared by incubating with the pre-washed beads for 20 min at RT

on a rotator. The pre-cleared lysates were incubated with anti-LSD1 (2139, CST) primary antibody overnight at 4°C on a rotator to form the immunocomplex. Control samples were incubated with an anti-rabbit IgG isotype control antibody (ABIN101961, Antibodies online). The protein–antibody complexes were then immunoprecipitated for 20 min at RT on a rotator with protein A sepharose magnetic beads which were pre-washed with 1× Cell Lysis Buffer. After incubation, bead-bound immune complexes were washed three times with 1× Cell Lysis Buffer followed by 10 PBS washes prior to MS analysis. Confirmation of successful co-IP was conducted by immunoblotting according to standard procedures.

## co-IP LC-MS/MS analysis

The nano-HPLC system used was an UltiMate 3000 RSLC nanosystem (Thermo Fisher Scientific, Amsterdam, Netherlands) coupled to a Q Exactive HF-X mass spectrometer (Thermo Fisher Scientific, Bremen, Germany), equipped with a Proxeon nanospray source (Thermo Fisher Scientific, Odense, Denmark). Peptides were loaded onto a trap column (Thermo Fisher Scientific, Amsterdam, Netherlands, PepMap C18, 5 mm × 300 μm ID, 5 μm particles, 100 Å pore size) at a flow rate of 25 μl/min using 0.1% TFA as mobile phase. After 10 min, the trap column was switched in line with the analytical column (Thermo Fisher Scientific, Amsterdam, Netherlands, PepMap C18, 500 mm × 75 μm ID, 2 μm, 100 Å). Peptides were eluted using a flow rate of 230 nl/min, and a binary 3-h gradient, respectively, 225 min.

The gradient starts with the mobile phases: 98% A (water/formic acid, 99.9/0.1, v/v) and 2% B (water/acetonitrile/formic acid, 19.92/80/0.08, v/v/v), increases to 35% B over the next 180 min, followed by a gradient in 5 min to 90% B, stays there for 5 min, and decreases in 2 min back to the gradient 98% A and 2% B for equilibration at 30°C.

The Q Exactive HF-X mass spectrometer was operated in data-dependent mode, using a full scan ($m/z$ range 380–1,500, nominal resolution of 60,000, target value 1E6) followed by 10 MS/MS scans of the 10 most abundant ions. MS/MS spectra were acquired using a normalized collision energy of 28, isolation width of 1.0 $m/z$, and resolution of 30,000, and the target value was set to 1E5. Precursor ions selected for fragmentation (exclude charge state 1, 7, 8, > 8) were put on a dynamic exclusion list for 60 s. Additionally, the minimum AGC target was set to 5E3 and the intensity threshold was calculated to be 4.8E4. The peptide match feature was set to preferred and the exclude isotopes feature was enabled.

## co-IP MS data processing

For peptide identification, the RAW files were loaded into Proteome Discoverer (version 2.3.0.523, Thermo Scientific). All hereby created MS/MS spectra were searched using MSAmanda v2.3.0.12368, Engine version v2.0.0.12368 (Dorfer et al, 2014). For the 1st step search, the RAW files were searched against the SwissProt database, taxonomy Homo sapiens (20.341 sequences; 11,361,548 residues), supplemented with common contaminants, using the following search parameters: The peptide mass tolerance was set to ± 5 ppm and the fragment mass tolerance to 15 ppm. The maximal number of missed cleavages was set to 2, using tryptic enzymatic specificity. The result was filtered to 1% FDR on protein

level using the Percolator algorithm (Käll *et al*, 2007) integrated in the Thermo Proteome Discoverer. A subdatabase was generated for further processing.

For the 2nd step, the RAW files were searched against the created subdatabase using the following search parameters: Beta-methylthiolation on cysteine was set as a fixed modification, oxidation on methionine, deamidation of asparagine and glutamine, acetylation on lysine, phosphorylation on serine, threonine, and tyrosine, methylation, and di-methylation on lysine and arginine, tri-methylation on lysine, ubiquitination on lysine, and biotinylation on lysine were set as variable modifications. Monoisotopic masses were searched within unrestricted protein masses for tryptic enzymatic specificity, respectively. The peptide mass tolerance was set to $\pm$ 5 ppm and the fragment mass tolerance to $\pm$ 15 ppm. The maximal number of missed cleavages was set to 2. The result was filtered to 1% FDR on protein level using the Percolator algorithm integrated in Proteome Discoverer. The localization of the post-translational modification sites within the peptides was performed with the tool ptmRS, based on the tool phosphoRS (Taus *et al*, 2011). Peptide areas were quantified using the in-house-developed tool apQuant (Doblmann *et al*, 2019).

### Nuclei isolation and TMT-labeling

Cells were pre-treated with either 1 μM GSK-LSD1 or DMSO 24 h prior to the nuclei isolation. $1 \times 10^6$ cells were washed with ice-cold PBS, resuspended in CE + NP40 buffer (10 mM HEPES, 60 mM KCl, 1 mM EDTA, 0.075% (v/v) NP-40, 1 mM DTT, pH 7.6.), and incubated for 3 min at 4°C. Nuclear and cytoplasmic fractions were separated by centrifugation for 4 min at 20,000 RCF at 4°C. The cytoplasmic supernatant was removed, and the cell nuclei were washed three times with CE buffer (10 mM HEPES, 60 mM KCl, 1 mM EDTA, 1 mM DTT, pH 7.6). Successful fractionation of cytoplasmic and nuclear fractions was confirmed by immunoblotting. Nuclear proteins were prepared for Mass Spectrometry Analysis using the iST-NHS kit (P.O.00030, PreOmics) together with the TMT10plex™ Isobaric Label Reagent Set (90110, Thermo Fisher Scientific) according to standard procedures.

### TMT LC-MS/MS analysis

The samples were resolved in 50 μl 0.1% FA, and 1% were analyzed on a monolithic HPLC column. The sample concentration was determined based on the UV traces, and equal amounts of each channel were mixed. Five hundred nanogram of the mixture sample and each single channel were subsequently analyzed in an LC-MS/MS experiment to determine the labeling efficiency and the mixing ratio by calculating the median value of the 500 most abundant proteins. Based on the results of the LC-MS/MS analysis, different amounts of each channel were added to correct the median value to 1:1: 1:1: 1:1 and ensure mixing of equal amounts of all samples. Five hundred nanogram of this sample was again analyzed in an LC-MS/MS experiment to confirm the bias. The mixed sample was lyophilized and dissolved in 50 μl SCX A Buffer (5 mM NaPO$_4$ pH 2.7, 15% ACN). SCX was performed using an Ultimate system (Thermo Fisher Scientific) at a flow rate of 35 μl/min and a TSKgel column (ToSOH, 5 μm particles, 1 mm i.d. × 300 mm). Fifty microgram of peptide was loaded on the column. For the separation, a

ternary gradient was used. Starting with 100% buffer A (5 mM NaPO$_4$ pH 2.7, 15% ACN) for 10 min, followed by a linear increase to 20% buffer B (5 mM NaPO$_4$ pH 2.7, 1 M NaCl, 15% ACN) and 50% buffer C (5 mM 5 mM NaPO$_4$ pH 6, 15% ACN) in 10 min, to 25% buffer B and 50% buffer C in 10 min, 50% buffer B, and 50% buffer C in 5 min and an isocratic elution for further 15 min. The flow-through was collected as a single fraction, and along the gradient, fractions were collected every minute from 10 min to 70 min and stored at −80°C.

### TMT-MS data processing

For peptide identification, the RAW files were loaded into Proteome Discoverer (version 2.3.0.523, Thermo Scientific). All hereby created MS/MS spectra were searched using MSAmanda v2.0.0.14114 (Dorfer *et al*, 2014). The RAW files were searched against the Uniprot database, using Homo sapiens as suborganism (20,541 sequences; 11,395,640 residues). The following search parameters were used: Acetylhypusine on cysteine was set as a fixed modification, oxidation on methionine, deamidation of asparagine and glutamine, phosphorylation on serine, threonine and tyrosine, 10-plex tandem mass tag® (TMT) on lysine, methylation on lysine and arginine, di-methylation on lysine and arginine, tri-methylation on lysine, acetylation on lysine, and Carbamylation and 10-plex TMT on peptide-N-term were set as variable modifications. Monoisotopic masses were searched within unrestricted protein masses for tryptic enzymatic specificity. The peptide mass tolerance was set to $\pm$ 5 ppm and the fragment mass tolerance to $\pm$ 15 ppm. The maximal number of missed cleavages was set to 2. The result was filtered to 1% FDR on protein level using the Percolator algorithm integrated with Thermo Proteome Discoverer. The localization of the modification sites within the peptides was performed with the tool ptmRS, which is based on phosphoRS (Taus *et al*, 2011). Peptides were quantified based on Reporter Ion intensities extracted by the "Reporter Ions Quantifier"-node implemented in Proteome Discoverer. Proteins were quantified by summing unique and razor peptides. Protein abundances-normalization was done using sum normalization. Statistical significance of differentially expressed proteins was determined using limma (Smyth, 2005).

### HMG20B protein analysis

To examine sequence conservation, homologous sequences were collected in NCBI blast searches starting with human HMG20B (UniProt: Q9P0W2; region 177–316) against the UniProt reference proteomes or NCBI non-redundant protein database and applying highly significant E-values (< 1e-10, (Altschul *et al*, 1997)). Sequences were selected for a wide taxonomic range, including paralogs, and aligned with mafft (version 7.427, L-INS-i method, (Katoh & Toh, 2008)), and the conservation score was calculated with AACon (version 1.1, Karlin method, http://www.compbio.dundee.ac.uk/aacon). Sequence alignments were visualized with Jalview (Waterhouse *et al*, 2009).

### HMG20B cDNA cloning and rescue experiment

cDNA of HMG20B was cloned into a lentiviral pTwist backbone encoding for SFFV-V5-cDNA-P2A-mCherry-IRES-Puro-WPRE. At the

location of cDNA, we either expressed the codon-optimized wild-type (WT) version of HMG20B or any of the codon-optimized HMG20B domain deletion mutants (ΔHMG, ΔALPHA, ΔCC). An empty vector was generated with IRFP670 at the position of cDNA. All vectors were generated by Twist Bioscience and validated by Sanger sequencing.

Cells were transduced to express either wild-type HMG20B, any of the HMG20B domain deletion mutants (ΔHMG, ΔALPHA, ΔCC) or empty vector (EV-IRFP670), and selected using 2 µg/ml puromycin. Next, cells were co-transduced with a doxycycline-inducible, shRNA-expressing backbone at a multiplicity of infection (MOI) of ~ 1 (T3G-GFP-miRE-PGK-Puro-IRES-rtTA3). shRNA were either targeting endogenous HMG20B or Renilla luciferase. The shRNA expression was induced with 500 ng/ml doxycycline. The abundance of transduced cells was followed by assessing the abundance of the GFP$^+$ mCherry$^+$ population over time by flow cytometry (FACS LSR Fortessa cytometer).

### HMG20B immunoprecipitation

Cells were transfected with plasmids encoding for V5-tagged wild-type HMG20B, any of the HMG20B domain deletion mutants (ΔHMG, ΔALPHA, ΔCC) or empty vector (EV - IRFP670) using polyethylenimine (PEI) as previously reported (Muhar et al, 2018). Cells were washed with ice-cold PBS and lysed on ice with 1× Cell Lysis Buffer (9803S, CST). Lysates were cleared by centrifugation at 14,000 g for 10 min at 4°C. Protein concentrations were determined with the Pierce™ BCA Protein Assay kit (23227, Thermo Fisher Scientific) according to the manufacturer's instructions. Lysates were incubated with α-V5 agarose beads (A7345, Sigma-Aldrich) for 24 h at 4°C and washed three times in 1× Cell Lysis Buffer with 300 mM NaCl. Confirmation of successful co-IP was conducted by immunoblotting according to standard procedures.

### Cell cycle staining

Cells were treated for indicated times with 100 nM GSK-LSD1 or DMSO and labeled for 3 h with 10 µM EdU in standard cell culture medium, and EdU was subsequently detected using the Click-iT™ Plus EdU Flow Cytometry Alexa Fluor™ 647 Assay Kit (C10634, Invitrogen) according to the manufacturer's instructions. EdU-labeled cells were stained for DNA content using the FxCycle™ PI/RNase (F10797, Invitrogen) kit for 30 min before acquiring 20,000 single-cell events per sample by flow cytometry (FACS LSR Fortessa cytometer). For Annexin V/DAPI staining, cells were washed twice with cold PBS and resuspended in Annexin V Binding Buffer (422201, BioLegend) to a concentration of $1 \times 10^5$ cells/100 µl. Five microliter FITC Annexin V (B640906, BioLegend) and 1 µl DAPI (1 µg/ml final concentration) were added, and samples were incubated for 15 min at RT in the dark before acquiring 20,000 single-cell events per sample by flow cytometry (FACS LSR Fortessa cytometer). For TMRE staining, cells were treated for up to 6 days with 100 nM GSK-LSD1 or DMSO. At a concentration of $5 \times 10^5$ cells/ml, cells were stained with 150 nM TMRE (final concentration) for 5 min at room temperature in the dark (Crowley et al, 2016), before acquiring 20,000 single-cell events per sample by flow cytometry (FACS LSR Fortessa cytometer).

### Drug wash-out experiment

Cells were pre-treated with either 100 nM GSK-LSD1 or DMSO. After 24 h, cells were spun down, washed, and resuspended in fresh medium without drugs. Cells were harvested at different time points. Proteins and RNA were extracted as described above. Immunoblot and qPCR analysis were performed according to standard procedures. EdU/PI and Annexin V/DAPI stainings with FACS read-out were performed, according to the procedure described above, at day 8 post-wash-out.

### PARP1 cleavage

Cells were treated either with ORY-1001 (10 nM or 100 nM), etoposide (10 µM, HY-13629, MedChemExpress), or DMSO for 24 h. Cells were harvested, washed with PBS, and lysed with RIPA buffer as described above. Immunoblotting was conducted according to standard procedures.

### Caspase-Glo

Cells were treated either with ORY-1001 (10 nM), GSK-LSD1 (10 nM), or DMSO for 6 days. As a positive control for apoptosis, cells were treated with etoposide (10 µM) for 24 h. Apoptosis was assessed with Caspase-Glo 3/7 assay (G8090, Promega) according to the manufacturer's instruction.

### Ki-67 staining

Cells were treated either with ORY-1001 (1 µM), GSK-LSD1 (1 µM), or DMSO for 3 days. Cells were harvested for 5 min at 400 g at RT and stained with Pacific Blue™ anti-human Ki-67 Antibody (B350511, BioLegend) according to eBioscience™ Transcription Factor Staining Buffer Set standard procedure. Twenty thousand single-cell events were acquired per sample on a FACS LSR Fortessa cytometer, and Pacific Blue™-positive fractions were determined with the FlowJo™10 software.

### H&E and immunofluorescence staining

Four-micron-thick sections were stained with hematoxylin (6765002, Thermo Scientific) and eosin (6766008, Thermo Scientific; H&E) for histological examination using the Thermo Scientific™ Gemini AS Automated Slide Stainer. Tissues for immunofluorescence (IF) staining of tumors were obtained after fixation in 4% PFA for 24 h and two-step tissue dehydration to 30% sucrose at 4°C. Tumors were sliced using a sledge microtome (20 µm), placed on slides, dried for 30 min, fixed with 4% PFA for 10 min at RT, and permeabilized with 0.2% Triton for 30 min at RT. Next, slides were blocked in 10% goat serum, 2% BSA, and 0.25% Triton in PBS for 1 h at room temperature (RT). Primary antibodies were incubated overnight in the blocking solution at 4°C and on the next day for 30 min at RT. The following antibodies against human epitopes were used: anti-Ki-67 (ab15580, Abcam, 1:200) and anti-cleaved-Casp-3 (9661, CST, 1:400). After five washes in PBS/0.25% Triton, secondary antibodies were added for 1 h on RT at a concentration of 1:1,000 followed by five washes. Then, nuclei were stained with Hoechst 33258 (H3569,

Invitrogen™), and tumor slices were washed with PBS, transferred onto glass slides, and mounted with Prolong™ Gold Antifade reagent (P36970, Thermo Fisher Scientific).

TUNEL staining for detection of apoptosis-induced double-strand breaks was performed using the TUNEL Apoptosis Detection Kit (A050, ABP Biosciences) according to the manufacturer's instructions with Alexa Fluor 555 (A-21428, Thermo Fisher Scientific). Images were collected with a Zeiss LSM800 Axio Imager microscope equipped with Zeiss ZEN BLUE software. For quantification of IF stainings, five regions of interest (ROI) were acquired per tumor from the tumor margin and counted manually using Fiji with the Cell Counter Plugin. The tumor margin was defined as a non-necrotic tumor border as assessed based on H&E staining. For each condition, ≥ 3 tumors with each ≥ 5 ROIs were counted.

## QUANTseq

Cells were treated for 24 h or 6 days with 100 nM GSK-LSD1 or DMSO. At harvest time points, cells were washed in PBS and cell pellets snap-frozen. RNA extraction was performed using an in-house total RNA magnetic beads-based purification protocol. 3′-end mRNA sequencing libraries (Quantseq) were prepared from 500 ng RNA using the QuantSeq 3′ mRNA-Seq Library Prep Kit FWD for Illumina (Lexogen) and PCR Add-on Kit for Illumina kits (020.96, Lexogen) according to the manufacturer's instructions. Sequencing was performed on an Illumina HiSeq 2500 in 50 bp single-end mode.

Analysis of QUANTseq data was performed with an in-house pipeline, briefly: Adapter and polyA sequences were clipped (bbdk, 38.06, https://sourceforge.net/projects/bbmap/), abundant sequences were removed (bbmap, 38.06; https://sourceforge.net/projects/bbmap/), and cleaned reads were aligned against the genome (hg38) with (STAR, 2.6.0c (Dobin et al, 2013)). Raw reads were mapped to 3′ UTR annotations of the same gene and collapsed to gene level by Entrez Gene ID with featureCounts (v1.6.2). Differentially expressed genes were calculated using DESeq2 (v1.18.1 (Love et al, 2014)). Pathway enrichment analysis was performed using gProfiler (Raudvere et al, 2019) and Metascape (Zhou et al, 2019). Gene set enrichment analysis (Subramanian et al, 2005) was performed using the GSEA tool (4.0.3) with a normalized gene expression table and standard parameters for enrichment testing. Gene-based motif analysis was performed using Homer (4.11) (Heinz et al, 2010) with the findMotifs.pl script and default settings.

## CUT&RUN

CUT&RUN was performed according to the high-calcium/low-salt digestion protocol described previously (Meers et al, 2019) together with the CUT&RUN Assay Kit (86652, CST). Briefly, cells were harvested, washed, and bound to activated Concanavalin A-coated magnetic beads and permeabilized. The bead–cell complex was incubated overnight with the respective antibody at 4°C. Cells were washed three times and resuspended in 100 μl pAG/MNase and incubated for 1 h at RT. DNA fragments were isolated by phenol/chloroform extraction followed by ethanol precipitation. CUT&RUN libraries were prepared using the NEBNext® Ultra™ II DNA Library Prep Kit for Illumina® (E7645, NEB) according to the

manufacturer's instructions. Library quality was assessed using a Fragment Analyzer system (Agilent). Sequencing was performed on an Illumina HiSeqV4 platform in 125 bp paired-end mode. Antibodies used are as follows: anti-rabbit IgG isotype (ABIN101961, Antibodies online, 1:50), anti-H3K4me1 (C15410194, Diagenode, 1:50), anti-H3K27ac (ab4729, Abcam, 1:50), and anti-H3K27me3 (9733, CST, 1:50).

CUT&RUN data were analyzed utilizing the nf-core ChIPSeq pipeline (Wang et al, 2019; Ewels et al, 2020). Data quality was checked by various QC modules (fastqc, samtools flagstat, samtools stats, samtools idxstats, cutadapt reports, preseq, picard MarkDuplicates, MACS2 reports, and phantompeakqualtools). Read mapping was performed with bwa mem v0.7.17 against the human genome assembly hg38 using the parameters -M. Peak calling was performed with MACS2 v2.1.2 with the following parameters: –broad –broad-cutoff 0.1 –keep-dup all with anti-rabbit IgG isotype used as control samples. Aligned reads of the individual replicates were merged using samtools v1.9 (Li et al, 2009), and RPGC-normalized tracks were calculated and plotted as composite density plots using deep-tools v3.1.2 (Ramírez et al, 2016).

## SLAMseq

4-thiouridine (4sU) toxicity was assessed by treating cells with 500 μM 4-thiouridine (T4509, Sigma-Aldrich) or with DMSO. Cell viability was read out with the CellTiter-Glo® 2.0 Cell Viability Assay (Promega) according to the manufacturer's instructions at treatment start ($T_0$) and after 1, 3, and 6 h. Cell viability at each time point was calculated relative to the respective measure at $T_0$. For labeling of nascent RNA, cells grown at about 50% of the maximum cell density counted on a hemocytometer were pre-treated with a GSK-LSD1 at 100 nM or DMSO for 30 min to pre-established full target inhibition. Newly synthesized RNA was labeled by the addition of 500 μM 4-thiouridine (4sU,) for 60 or 360 min, respectively. Cells were harvested and immediately snap-frozen. RNA extraction was performed using an in-house magnetic bead-based purification protocol as mentioned above. As previously reported (Muhar et al, 2018), alkylation of total RNA was induced with 10 mM iodoacetamide (I1149-5G, Sigma-Aldrich) for 15 min and RNA was repurified by ethanol precipitation. As input for generating 3′-end mRNA sequencing (Quantseq) libraries, 500 ng alkylated RNA was used and prepared with a commercially available kit (QuantSeq 3′ mRNA-Seq Library Prep Kit FWD for Illumina and PCR Add-on Kit for Illumina, Lexogen). Sequencing was performed on an Illumina NovaSeq SP platform in 100 bp single-end mode.

Data analysis was performed as previously reported (Muhar et al, 2018). Briefly, gene and 3′ UTR annotations for hg38 were obtained from UCSC table browser. Adapters were trimmed from raw reads using cutadapt through the trim_galore (v0.3.7) wrapper tool with adapter overlaps set to 3 bp for trimming. Trimmed reads were further processed using SlamDunk (Neumann et al, 2019) using the (slamdunk all) method. Reads were filtered for having ≥ 2 T>C conversions. Differentially expressed genes were calculated using DESeq2 (v1.18.1 (Love et al, 2014)). Pathway enrichment analysis was performed using gProfiler (Raudvere et al, 2019) and Metascape (Zhou et al, 2019).

## ATACseq

ATACseq was performed according to the Omni-ATACseq protocol (Corces et al, 2017). Briefly, 100,000 cells were pelleted and resuspended in resuspending buffer (10 mM Tris–HCl, 10 mM NaCl, 3 mM MgCl2, 0.1% Nonidet P40, 0.1% Tween-20, and 0.01% digitonin). After nuclei isolation, cells were resuspended in TD buffer with in-house purified Tn5 transposase. The transposition reaction was performed at 37°C for 30 min. DNA fragments were purified from the reaction using a DNA Clean & Concentrator-5 kit (Zymo Research) and amplified using the NEBNext High Fidelity PCR Mix (NEB). Library quality was assessed using a Fragment Analyzer system (Agilent). ATACseq libraries were quantified using a KAPA library quantification kit and sequenced on an Illumina NextSeq550 instrument in 75 bp paired-end mode.

ATACseq data were analyzed using the nf-core/ATACseq pipeline (Patel et al, 2019) aligning against the human genome hg38 and calling peaks with MACS2 in narrow- and broad-peak mode. Aligned reads of the individual replicates were merged using samtools v1.9 (Li et al, 2009), and RPGC-normalized tracks were calculated and plotted as composite density plots using deeptools v3.1.2 (Ramírez et al, 2016).

## TGFβ signaling pathway induction

Cells were pre-treated with either active BMP2 recombinant protein (10 ng/ml, 355-BM-010, Bio-Techne Ltd), active BMP4 recombinant protein (10 ng/ml, 120-05-ET, PeproTech), ORY-1001 (1 μM), GSK-LSD1 (1 μM), or DMSO for 24 h. Prior to protein extraction, each condition was split and was either co-treated with the proteasome inhibitor MG-132 (1 μM, S2619, Selleck Chemicals) or DMSO for 2 h. Cells were pelleted, washed with PBS, and lysed with RIPA buffer as described above. Immunoblotting was conducted according to standard procedures.

## In vivo tumor treatment

PeTa cells ($2 \times 10^6$) mixed with Matrigel (3562377, Corning) were injected subcutaneously into the left flank of 5- to 8-week-old sex-randomized NSG mice. NSG (NOD-scid IL2Rgamma$^{null}$) mice were purchased from The Jackson Laboratory (strain 005557). Prior to tumor cell injection, the mice were injected intraperitoneally with analgesic ketamine (100 mg/kg), xylazine (10 mg/kg), and acepromazine (3 mg/kg). Animal weight was frequently monitored. Tumor size was monitored by tumor measurement with an electronic caliper. Measured tumor size was determined by the formula $[x \ (mm)^2 * y \ (mm)]/2 = $ volume $(mm^3)$, where $x$ refers to the shortest measurement and $y$ the longest. Once tumor size reached 50 mm$^3$ (day 22 post-tumor injection), mice were either treated with GSK-LSD1 (1 mg/kg) or vehicle (0.9% saline) by intraperitoneal (IP) injection. Mice were treated during five consecutive days followed by 2 days off-treatment (5 days ON/2 days OFF) repeatedly. Mice were sacrificed by cervical dislocation when either total tumor volume was larger than 1,500 mm$^3$ or when one of the measured lengths was larger than 1,500 mm. At least three mice were sacrificed after 1 day (D1) or 10 days (D10) of treatment to assess the early effects of LSD1i. All experiments using animals were performed in accordance with our protocol approved by the

**The paper explained**

**Problem**
Merkel cell carcinoma (MCC) is a rare skin cancer with limited treatment options. Immunotherapy has proven to be effective only in a subgroup of patients. There is therefore an unmet need to identify new entry points for targeted therapies to treat MCC patients.

**Results**
In this study, we identify the lysine-specific histone demethylase 1A (LSD1/KDM1A) as a strong genetic and pharmacological dependency in MCC. Inhibition of LSD1 impairs the integrity of the LSD1-CoREST complex, leading to displacement and degradation of HMG20B, an essential subunit of this complex. LSD1 inhibition induces the derepression of key regulators of the neuronal lineage, promotes cell death, and strongly inhibits MCC cell growth in vitro and in vivo.

**Impact**
Our study reveals the importance of LSD1 for proliferation and maintenance of cell identity in MCC and suggests that differentiation therapies that have been shown to be effective in leukemia may be a powerful therapeutic option for treating MCC. There is growing evidence that cancer cells exploit cellular plasticity and dedifferentiation programs to escape destruction by the immune system. The combination of LSD1 inhibitors with checkpoint inhibitors could therefore represent a promising treatment strategy for MCC patients.

Austrian Ministry (BMBWF-66.015/0009-V/3b/2019 or GZ: 340118/2017/25). Kaplan–Meier survival curves were plotted with the software GraphPad PRISM 8 and significance determined by log-rank Mantel–Cox test. The significance of the grouped xenograft growth was determined by Student's t-test or Mann–Whitney for non-normal distributed data.

## In vivo micrometastatic treatment

PeTa cells ($2 \times 10^6$) mixed with Matrigel were injected subcutaneously into the left flank of 5- to 8-week-old sex-randomized NSG mice. NSG (NOD-scid IL2Rgamma$^{null}$) mice were purchased from The Jackson Laboratory (strain 005557). Prior tumor injection, the mice were injected intraperitoneally with analgesic ketamine (100 mg/kg), xylazine (10 mg/kg), and acepromazine (3 mg/kg). Animal weight was frequently monitored. Tumor size was monitored by tumor measurement with an electronic caliper. Measured tumor size was determined by the formula $[x \ (mm)^2 * y \ (mm)]/2 = $ volume $(mm^3)$, where $x$ refers to the shortest measurement and $y$ the longest. Mice were either treated with GSK-LSD1 (1 mg/kg), ORY-1001 (0.03 mg/kg), or vehicle (0.9% saline) by intraperitoneal (IP) injection from day 1 after tumor injection on. Mice were treated during five consecutive days followed by 2 days off-treatment (5 days ON/2 days OFF) repeatedly. Mice were sacrificed by cervical dislocation when either total tumor volume was larger than 1,500 mm$^3$ or when one of the measured lengths was larger than 1,500 mm. All experiments using animals were performed in accordance with our protocol approved by the Austrian Ministry (BMBWF-66.015/0009-V/3b/2019 or GZ: 340118/2017/25). Survival curves were plotted with the software GraphPad PRISM 8 and significance determined by the log-rank Mantel–Cox test. The significance of the

grouped xenograft growth was determined by Student's $t$-test or Mann–Whitney for non-normal distributed data.

### In vivo pre-treatment experiment

PeTa cells were pre-treated with DMSO or GSK-LSD1 (100 nM) 1 day, 3 days or 6 days prior cell injection. Cells ($2 \times 10^6$) mixed with Matrigel were injected subcutaneously into the left flank of 5- to 8-week-old male and female NSG mice. NSG (NOD-*scid* IL2Rgamma$^{null}$) mice were purchased from The Jackson Laboratory (strain 005557). Prior tumor injection, the mice were injected intraperitoneally with analgesic ketamine (100 mg/kg), xylazine (10 mg/kg), and acepromazine (3 mg/kg). Animal weight was frequently monitored. Tumor size was monitored by tumor measurement with an electronic caliper. Measured tumor size was determined by the formula $[x \ (mm)^2 \ * \ y \ (mm)]/2 =$ volume (mm$^3$), where $x$ refers to the shortest measurement and $y$ the longest. Mice were sacrificed by cervical dislocation when either total tumor volume was larger than 1,500 mm$^3$ or when one of the measured lengths was larger than 1,500 mm or if mice showed signs of unhealthiness. All experiments using animals were performed in accordance with our protocol approved by the Austrian Ministry (BMBWF-66.015/0009-V/3b/2019 or GZ: 340118/2017/25). Survival curves were plotted with the software GraphPad PRISM 8 and significance determined by the log-rank Mantel–Cox test. The significance of the grouped xenograft growth was determined by two-way ANOVA.

### Merkel cell isolation and SMARTseq

Dorsal skin was removed from euthanized 4-week-old male and female C57BL/6 mice from IMP in-house breeding facility and dissociated with the Epidermis Dissociation Kit (Miltenyi Biotec, Cat. 130-095-928) according to the manufacturer's instructions. Briefly, the dorsal skin patches were first incubated with enzyme G to dissociate the epidermis from the dermis. Next, the epidermis was dissociated into a single-cell suspension with the gentle-MACS™ Dissociator and enzymes P and A. The Merkel cell surface marker NCAM (neural cell adhesion molecule, CD56) was used for the enrichment of a Merkel cell population. The NCAM$^+$ (CD56$^+$) Merkel cells in our cell suspension were bound by anti-PSA-NCAM microbeads (130-097-859, Miltenyi Biotec) and extracted with the MACS® Column system. Then, the Merkel cell-enriched population was centrifuged for 5 min at 400 $g$ and resuspended in FACS buffer (PBS supplemented with 0.5% bovine serum albumin [BSA] and 2 mM EDTA). Merkel cells were stained with anti-PSA-NCAM-PE antibody (130-117-394, Miltenyi Biotec) or with mouse IgM-PE isotype control (130-120-070, Miltenyi Biotec) according to manufacturer's instructions, for 10 min at 4°C. Cells were washed with FACS buffer and centrifuged for 5 min at 4°C. Next, to exclude dead cells, Merkel cells were resuspended in FACS buffer and stained with DAPI (1 μg/ml) for 30 min at 4°C. NCAM-PE$^+$ DAPI$^-$ Merkel cells, as well as bulk epidermis cells (DAPI$^-$), were FACS-sorted in triplicates, each with 500 events in triplicates in a 96-well plate. The gating strategy for the FACS sorting is depicted in Appendix Fig S2. Smart-seq2 (Picelli *et al*, 2013) libraries were prepared and sequenced on an Illumina HiSeqV4 in 50 bp single-end mode.

Analysis of SMARTseq data was performed with an in-house pipeline, briefly: Adapter and polyA sequences were clipped (bbdk, 38.06, https://sourceforge.net/projects/bbmap/), abundant sequences were removed (bbmap, 38.06, https://sourceforge.net/projects/bbmap/), and cleaned reads were aligned against the GRCm38 genome (STAR, 2.6.0c, (Dobin *et al*, 2013)). Aligned reads were counted with featureCounts (v1.6.2). Differentially expressed genes were calculated using DESeq2 (v1.18.1, (Love *et al*, 2014)).

## Data availability

The datasets produced in this study are available in the following databases:

- QUANTseq data: Gene Expression Omnibus: GSE147815 (https://www.ncbi.nlm.nih.gov/geo/query/acc.cgi?acc = GSE147815)
- SLAMseq data: Gene Expression Omnibus: GSE147816 (https://www.ncbi.nlm.nih.gov/geo/query/acc.cgi?acc = GSE147816)
- CUT&RUN data: Gene Expression Omnibus: GSE148103 (https://www.ncbi.nlm.nih.gov/geo/query/acc.cgi?acc = GSE148103)
- ATACseq data: Gene Expression Omnibus: GSE147814 (https://www.ncbi.nlm.nih.gov/geo/query/acc.cgi?acc = GSE147814)
- SMARTseq data: Gene Expression Omnibus: GSE148102 (https://www.ncbi.nlm.nih.gov/geo/query/acc.cgi?acc = GSE148102)
- Mass spectrometry data: ProteomeXchange: PXD020590 (http://www.ebi.ac.uk/pride/archive/projects/PXD020590).

Expanded View for this article is available online.

### Acknowledgements

We thank all members of the Obenauf lab for experimental support and discussions. We further thank the Zuber lab for providing reagents and technical help as well as all core facilities of IMP/IMBA, especially the Histopathology, BioOptics, Molecular Biology Service and Translational Medicine facilities. We thank the Protein Chemistry facility for their mass spectrometry expertise and experimental support, especially Susanne Opravil, Richard Imre, Michael Schutzbier, and Karel Stejskal. The Epigenetics Probes Collection was supplied by the Structural Genomics Consortium (www.thesgc.org) under an Open Science Trust Agreement. We thank Dr. David Schrama (University Wuerzburg, Germany) for providing the MCC cell lines used in this study. This work was funded by the Starting Grant of the European Research Council (ERC-StG-759590 to A.C.O) and the Vienna Science and Technology fund (#LS16-063 to A.C.O and T.W.). Research at the IMP is generously supported by Boehringer Ingelheim.

### Author contributions

LL, PSJ, ACO, and TW conceived the study, designed the experiments, interpreted the results, and wrote the manuscript. LL and PSJ equally contributed to *in vitro* and *in vivo* experiments and computational analysis of the data. IK performed immune-fluorescence stainings and quantifications. TN provided computational analysis support. AS performed protein sequence analysis. KM and his team led and analyzed the mass spectrometry experiments. ACO and TW jointly supervised the study. All authors read and approved the manuscript.

### Conflict of interests

The authors declare that they have no conflict of interest.

## For more information

Obenauf lab website: http://www.obenauflab.com/

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
