## [Review Process File · EMBO Molecular Medicine]

LSD1 inhibition induces differentiation and cell death in Merkel cell carcinoma

Lukas Leiendecker, Pauline Jung, Izabela Krecioch, Tobias Neumann, Alexander Schleiffer, Karl Mechtler, Thomas Wiesner, and Anna Obenauf

DOI: [10.15252/emmm.202012525](https://doi.org/10.15252/emmm.202012525)

Corresponding authors: Anna Obenauf (anna.obenauf@imp.ac.at) , Thomas Wiesner (thomas.wiesner@meduniwien.ac.at)

Review Timeline:

Submission Date:	14th Apr 20
Editorial Decision:	5th May 20
Revision Received:	5th Aug 20
Editorial Decision:	25th Aug 20
Revision Received:	7th Sep 20
Accepted:	8th Sep 20

Editor: Lise Roth

Transaction Report:

5th May 2020

Dear Anna,

Thank you for the submission of your manuscript to EMBO Molecular Medicine. We have now received feedback from the three reviewers who agreed to evaluate your manuscript. As you will see from the reports below, the referees acknowledge the interest of the study and are overall supporting publication of your work pending appropriate revisions.

A cross-commenting exercise helped defining more precisely the issues that should be addressed in priority:

- 1/ Strengthen the mechanistic analysis of the effect of LSD1 inhibition on MCC growth (in vitro and in vivo)
- 2/ Characterize the shift in cell fate induced by LSD1 inhibitor
- 3/ Analyze the effects of combination treatments
- 4/ Discuss the Nature Cell Biology paper reporting partly overlapping results.

Moreover, we will not ask you to address the impact of LSD1 inhibition in an immunocompetent MCC mouse model (due to the lack of model), or to repeat experiments in primary cells and PDX (due to the rareness of the disease and availability of patient material). Similarly, we do not expect results in other neuroendocrine tumors.

Apart from these specific points (immunocompetent mice, primary cells and other neuroendocrine tumors), addressing the reviewers' concerns in full will be necessary for further considering the manuscript in our journal, and acceptance of the manuscript will entail a second round of review. EMBO Molecular Medicine encourages a single round of revision only and therefore, acceptance or rejection of the manuscript will depend on the completeness of your responses included in the next, final version of the manuscript. For this reason, and to save you from any frustrations in the end, I would strongly advise against returning an incomplete revision.

When submitting your revised manuscript, please carefully review the instructions that follow below. Failure to include requested items will delay the evaluation of your revision:

- 1) A .docx formatted version of the manuscript text (including legends for main figures, EV figures and tables). Please make sure that the changes are highlighted to be clearly visible.
- 2) Individual production quality figure files as .eps, .tif, .jpg (one file per figure).
- 3) A .docx formatted letter INCLUDING the reviewers' reports and your detailed point-by-point responses to their comments. As part of the EMBO Press transparent editorial process, the point-by-point response is part of the Review Process File (RPF), which will be published alongside your paper.
- 4) A complete author checklist, which you can download from our author guidelines

(<https://www.embopress.org/page/journal/17574684/authorguide#submissionofrevisions>). Please insert information in the checklist that is also reflected in the manuscript. The completed author checklist will also be part of the RPF.

6) Before submitting your revision, primary datasets produced in this study need to be deposited in an appropriate public database (see <https://www.embopress.org/page/journal/17574684/authorguide#dataavailability>). Please remember to provide a reviewer password if the datasets are not yet public. The accession numbers and database should be listed in a formal "Data Availability" section (placed after Materials & Method). Please note that the Data Availability Section is restricted to new primary data that are part of this study. If this section does not apply to your study, please indicate: "This study includes no data deposited in external repositories"

7) We would also encourage you to include the source data for figure panels that show essential data. Numerical data should be provided as individual .xls or .csv files (including a tab describing the data). For blots or microscopy, uncropped images should be submitted (using a zip archive if multiple images need to be supplied for one panel). Additional information on source data and instruction on how to label the files are available at .

8) Our journal encourages inclusion of *data citations in the reference list* to directly cite datasets that were re-used and obtained from public databases. Data citations in the article text are distinct from normal bibliographical citations and should directly link to the database records from which the data can be accessed. In the main text, data citations are formatted as follows: "Data ref: Smith et al, 2001" or "Data ref: NCBI Sequence Read Archive PRJNA342805, 2017". In the Reference list, data citations must be labeled with "[DATASET]". A data reference must provide the database name, accession number/identifiers and a resolvable link to the landing page from which the data can be accessed at the end of the reference. Further instructions are available at .

9) We replaced Supplementary Information with Expanded View (EV) Figures and Tables that are collapsible/expandable online. A maximum of 5 EV Figures can be typeset. EV Figures should be cited as 'Figure EV1, Figure EV2' etc... in the text and their respective legends should be included in the main text after the legends of regular figures.

- Additional Tables/Datasets should be labeled and referred to as Table EV1, Dataset EV1, etc. Legends have to be provided in a separate tab in case of .xls files. Alternatively, the legend can be supplied as a separate text file (README) and zipped together with the Table/Dataset file. See detailed instructions here:

10) The paper explained: EMBO Molecular Medicine articles are accompanied by a summary of the articles to emphasize the major findings in the paper and their medical implications for the non-specialist reader. Please provide a draft summary of your article highlighting

11) For more information: There is space at the end of each article to list relevant web links for further consultation by our readers. Could you identify some relevant ones and provide such information as well? Some examples are patient associations, relevant databases, OMIM/proteins/genes links, author's websites, etc...

12) Every published paper now includes a 'Synopsis' to further enhance discoverability. Synopses are displayed on the journal webpage and are freely accessible to all readers. They include a short stand first (maximum of 300 characters, including space) as well as 2-5 one-sentences bullet points that summarizes the paper. Please write the bullet points to summarize the key NEW findings. They should be designed to be complementary to the abstract - i.e. not repeat the same text. We encourage inclusion of key acronyms and quantitative information (maximum of 30 words / bullet point). Please use the passive voice. Please attach these in a separate file or send them by email, we will incorporate them accordingly.

Please also suggest a striking image or visual abstract to illustrate your article. If you do please provide a png file 550 px-wide x 400-px high.

13) As part of the EMBO Publications transparent editorial process initiative (see our Editorial at <http://embomolmed.embopress.org/content/2/9/329>), EMBO Molecular Medicine will publish online a Review Process File (RPF) to accompany accepted manuscripts.

In the event of acceptance, this file will be published in conjunction with your paper and will include the anonymous referee reports, your point-by-point response and all pertinent correspondence relating to the manuscript. Let us know whether you agree with the publication of the RPF and as here, if you want to remove or not any figures from it prior to publication.

I look forward to receiving your revised manuscript.

With my best wishes,

Lise

Lise Roth, PhD
Editor
EMBO Molecular Medicine

To submit your manuscript, please follow this link:

Link Not Available

The system will prompt you to fill in your funding and payment information. This will allow Wiley to send you a quote for the article processing charge (APC) in case of acceptance. This quote takes into account any reduction or fee waivers that you may be eligible for. Authors do not need to pay any fees before their manuscript is accepted and transferred to our publisher.

Photos 400-800 DPI

*Additional important information regarding figures and illustrations can be found at <http://bit.ly/EMBOPressFigurePreparationGuideline>

***** Reviewer's comments *****

Referee #1 (Remarks for Author):

In the present manuscript, the authors performed a pharmacological screen in Merkel cell carcinoma cells, targeting epigenetic regulators. They discovered that LSD1 is required for MCC growth in vitro and in vivo and that HMG20B (BRAF35), a subunit of the LSD1-CoREST complex, is also essential

for MCC proliferation. LSD1 inhibition in MCC disrupts the LSD1-CoREST complex, directly activates TGF β signaling which induces the expression of regulators of the neuronal lineage, and activates a gene expression signature corresponding to normal Merkel cells. These results may offer a rationale for the use of LSD1 inhibitors against MCC.

The manuscript is well written and might, likely in short term, represent a real therapeutic strategy against Merkel cell carcinomas

Main Points

1. The authors analyze genome-wide RNAi screening data derived from the DepMap project, which includes genetic vulnerability maps for the MCC cell lines PeTa, MKL-1, and MKL-2. We examined the genes encoding the epigenetic regulators from their initial screen and found that LSD1 scores among the top 5 dependencies for MCC proliferation (Fig. 1e and Extended Data Fig. 1f). Would the finding of LSDi applicability apply only to MCC cells or in general there would be a potential application for other neuroendocrine cancers? If neuroendocrine cancer cells different from MCC would be analysed, would LSD1 display a high score for dependencies for proliferation?
2. If possible, a part of the data should be recapitulated in primary MCC cells (if available), at least for response to LSDi and their anticancer potential ex vivo or in PDX.
3. In figure 1e the dependency plot depicting the mean dependency of the 3 MCC cell lines PeTa, MKL-1, MKL-2 on the genes targeted by the compound library in Fig. 1a is shown. Would a combo scheme of LSD1i+BRD4i or other molecules targeting PRMT5, TAF1, WDR5 be more effective against MCC proliferation maintaining a similar specificity?
4. All the in vivo experiments have been performed in immune-compromised mice. If at all possible, the authors might include in vivo data in mice, which display a functional immune system. Alternatively, if technically this is not possible, the authors might comment on this issue in the discussion.
5. The authors identified core members, including the histone deacetylase HDAC2, and RCOR1, RCOR2, RCOR3, which serve as a scaffold for complex assembly, as well as non-canonical members, including GSE1, HMG20A, HMG20B (Fig. 3c and Extended Data Fig. 3a). Would a combo treatment targeting HDAC2 and LSD1 display any advantages against LSDi as monotherapy?
6. The authors found that LSD1i treatment of MCC leads to dissociation of this complex, in line with studies in leukemias (Fig. 3d,ne). This evaluation is performed at 24h. Following, the authors to investigate the molecular programs associated with LSD1i mediated growth inhibition, performed RNAseq of DMSO- and GSK-LSD1-treated MCC cells after 6 days. The promoter motifs in the deregulated genes had Co-/Rest binding motif enriched in the promoters of genes that become up-regulated in MCCs upon inhibition of LSD1, indicating a direct regulation of these genes by the LSD1-CoREST complex (Fig. 4d). The authors thus suggest that the therapeutic effect of LSD1i treatment is mediated by the expression of neuronal genes and differentiating MCC cells into a neuronal lineage. Interestingly, HMG20B, was identified as a subunit of the LSD1-CoREST complex in MCC, required to maintain full repression of neuron-specific genes. Given that at 24 hours the authors see a change in the LSD1-CoREST co-repressor complex, which complex they think is binding to the responsive promoters at d6? And how the time d6 was chosen?
7. By mapping the direct transcriptional response to LSD1i the authors identify TGF β signaling mediated activation of neuronal genes, suggesting that transcriptional activation of TGF β signaling in LSD1i-treated MCCs phenocopies receptor-ligand based TGF β activation and induces neuronal differentiation. A plethora of small molecules targeting inhibition of TGF β pathway, inhibition of BMP receptors, SMADs etc are commercially available and their use (with potential interference or synergy depending on the molecule chosen) in these settings might further prove the authors hypothesis even potentially suggesting the role of the different partners as well as additional therapeutic options.

Minor Points

The authors might enlarge the LSDi tested in MCC cell lines to a broader panel including some of them such as TCP, IMG-7289, INCB059872, CC-90011, and ORY-2001

Referee #2 (Remarks for Author):

Review manuscript EMBO Molecular Medicine- EMM-2020-12525

" LSD1 inhibitors induce neuronal differentiation of Merkel cell carcinoma"

The manuscript from Dr. Obenaus and colleagues entitled "LSD1 inhibitors induce neuronal differentiation of Merkel cell carcinoma" describes a vulnerability of Merkel Cell Carcinoma (MCC) cells to pharmacological treatment with inhibitors of the histone demethylase LSD1, unraveling part of the downstream events leading to differentiation of the MCC cells.

Specifically, the authors perform a pharmacological screen in MCC cells discovering a high sensitivity for LSD1 inhibitors treatment, and further consolidate this finding with knockdown experiments and data mining. The conclusion that MCC cells are dependent from LSD1 activity for their in vitro proliferation is then translated in vivo, showing that LSD1 inhibition affects tumor maintenance and tumor establishment in MCC xenograft models. The authors proceed to identify via IP-Mass Spectrometry interactors of LSD1, suggesting a prominent function of RCOR1 and of the non-canonical BRAF-histone deacetylase complex member HMG20B. Finally, the authors investigate via gene expression profiling and cell biology techniques the events downstream of LSD1 inhibition, identifying a function of LSD1 in inhibiting neuronal differentiation genes in MCC cells. This signature is overlapping with the expression profile of normal Merkel cells, and overall the data hint at the fact that LSD1 inhibition causes the de-repression of neuronal differentiation genes, thereby triggering a shift in cell fate that induces MCC to become normal Merkel cells. Finally, by profiling the newly transcribed RNAs with SLAMseq at different time points after LSD1 inhibition, the authors identify members downstream components of BMP signaling as bona fide effectors of the neuronal differentiation induction following LSD1 inhibition.

The work is well presented, technically sound and well controlled. The lack of treatment options for Merkel Cell Carcinoma makes the findings very interesting in terms of potential clinical applications, and the link between the LSD1 inhibitors treatment and the induction of physiological differentiation is very relevant. Part of the findings of this paper is overlapping with the recently published paper by Park and colleagues ("Merkel cell polyomavirus activates LSD1-mediated blockade of non-canonical BAF to regulate transformation and tumorigenesis", Nature Cell Biology 2020). Considering that the paper was published concomitantly of the submission, the non-scooping policy of EMBO related journal should apply to this case.

Following are some general recommendations that would improve the quality of the manuscript:

- The authors claim that LSD1 inhibitor treatment induces arrest of MCC cells growth in vitro and of tumor growth in vivo. The conclusion is that this is not due to cell death, but rather a decrease in the speed of the cell cycle due to the induction of a differentiation program.

For the in vitro experiments, this conclusion is supported (in Figure 4 and Extended Data Figure 4)

by morphological changes of the cells, changes in gene expression, caspase dependent apoptosis assays (PARP cleavage and Caspase cleavage) and KI-67 expression. Since this is a major point of the paper, the authors should rule out the possibility of other forms of caspase independent cell death (for instance, by performing TUNEL assay or AnnexinV staining) to prove that the decrease in cell number is really not due to any contribution of cell death. Also, concerning the cell cycle analysis, it would be advisable to extend it by performing BrdU/EdU labeling experiments, in order to describe in detail if the cell cycle is slowed down or arrested. Finally, concerning the induction of differentiation, I would like to have a validation of the expression of some Merkel cells markers by immunofluorescence, to prove that LSD1 inhibition really results in a change in cell identity.

The same concepts apply to the in vivo experiments described in Figure 2 and in Extended Data Fig. 2. In the plot in Fig. 2B, it seems that some tumors achieve a "steady state" size, some others have a slight increase, others a small decrease. In Ext. Data Fig.2A, which focuses on the early time points after the treatment, the impression is that overall tumors are still slowly growing for some time. It would be interesting to perform a histological analysis of the tumors at different time points (e.g. few days after the start of the treatment, at mid-term and at a later time point) to verify what is happening in the tumor mass. Are the cells switching completely identity and undergoing terminal differentiation? Is a part of the tumor undergoing cell death? Is there a significant remodeling of the microenvironment that could explain the slowdown in tumor growth? The effect on tumor maintenance and establishment is one of the most intriguing findings of this paper, and should definitely be analyzed in more detailed manner.

- To make the work more complete, the authors should assess the relative importance of the genes that are induced after LSD1 inhibition, for example by trying to mimic the differentiation induction upon genetic manipulation in the absence of LSD1 inhibition, e.g. by inducing ectopic expression of some of the factors that are implicated in Merkel cells normal differentiation (. The alternative approach would be showing the absence of differentiation induction after LSD1 inhibitors treatment upon knockdown of one or more of the genes implicated in the neuronal differentiation program.

- Also, it would be interesting to see how stable the shift in cell fate is. For instance, are MCC cells pre-treated with LSD1 inhibitor able to form tumors (so not at day1 after grafting, but at day -2 or -3)? If not (which would be my expectation), how long it takes for them to revert to a pro-tumorigenic phenotype, if this is at all possible, and what are the downstream events that are involved (e.g. it's merely a reactivation of LSD1?)

- In connection to this, is it conceivable that following LSD1 inhibitor treatment other epigenetic regulation complexes could take over the repression of pro-differentiation genes? Did the authors observe any upregulation of other demethylases that could suggest the acquisition of resistance mechanisms?

- My final remark is about the paper by Park et al. Obenaus and colleagues refer to this paper very briefly stating that "Interestingly, it was recently reported that the small T antigen of the Merkel cell polyomavirus establishes a dependency on LSD1 in MCC". The actual points of cross-talk between the work done by the two groups are quite numerous, and it would be a good idea to refer to the other paper a bit more extensively, even possibly by making use of the data when useful (e.g. of the ChIP-seq data).

It is a very thorough study with many different methods applied. The novelty is, however, limited since a recent publication demonstrated that Merkel cell polyomavirus encoded small T Antigen activates LSD-1 expression, and that LSD1 inhibition has an effect on MCC growth in vitro and in vivo.

Referee #3 (Remarks for Author):

In the manuscript of Leiendecker et al the authors performed a pharmacological screen in Merkel cell carcinoma (MCC) cells targeting epigenetic regulators. This analysis revealed that lysine-specific histone demethylase 1A (LSD1/KDM1A) is required for MCC growth. They further demonstrate that LSD1 inhibition triggers the TGF β signaling pathway resulting in the expression of key regulators of the neuronal lineage and a Merkel cell like gene signature.

As the authors pointed out, currently only immune checkpoint inhibitors are approved for therapy of metastatic patients, which sometimes have limited efficiency. Thus, there is still a medical need for efficient therapies for MCC. Since in virus-positive tumors no prominent mutations are present, targeted therapies directed against epigenetic regulators appear as a promising alternative. Therefore, the object of the study is absolutely justified. The authors present a well-written, thorough and experimental sound study. Their results are in line with a recent publication by Park et al (Nat Cell Biol., 2020) in which the authors demonstrate that Merkel cell polyomavirus (MCPyV) encoded small T antigen (sT) activates the expression of LSD1 rendering MCC cells sensitive to LSD1 inhibition.

Points, which should be addressed by the authors:

- For their analyses they only used MCPyV-positive MCC cells. The effect on virus-negative cells has not been tested. Therefore, the authors should make it clear that their observation only applies to the virus-positive subgroup.
- The presentation of the dependency score as violin and box plots for groups with only a few data points (three in the case of MCC) seems inappropriate.
- The authors couldn't detect apoptotic death upon LSD1 inhibitor treatment. Accordingly, the mechanism how LSD1 inhibition affects MCC cell growth is not clear. Probably a mixture between cell death and cell cycle arrest. To scrutinize the effect, the authors should perform cell counting and cell cycle analysis with Edu/Brdu staining.
- According to figure 1b even at highest doses of inhibitor the viability stays at about 25%. What happens in long time culture experiments? Do they become drug resistant?

Editorial Comments:

Thank you for the submission of your manuscript to EMBO Molecular Medicine. We have now received feedback from the three reviewers who agreed to evaluate your manuscript. As you will see from the reports below, the referees acknowledge the interest of the study and are overall supporting publication of your work pending appropriate revisions.

A cross-commenting exercise helped defining more precisely the issues that should be addressed in priority:

- 1/ Strengthen the mechanistic analysis of the effect of LSD1 inhibition on MCC growth (*in vitro* and *in vivo*)
- 2/ Characterize the shift in cell fate induced by LSD1 inhibitor
- 3/ Analyze the effects of combination treatments
- 4/ Discuss the Nature Cell Biology paper reporting partly overlapping results.

Moreover, we will not ask you to address the impact of LSD1 inhibition in an immunocompetent MCC mouse model (due to the lack of model), or to repeat experiments in primary cells and PDX (due to the rareness of the disease and availability of patient material). Similarly, we do not expect results in other neuroendocrine tumors.

Apart from these specific points (immunocompetent mice, primary cells and other neuroendocrine tumors), addressing the reviewers' concerns in full will be necessary for further considering the manuscript in our journal, and acceptance of the manuscript will entail a second round of review. EMBO Molecular Medicine encourages a single round of revision only and therefore, acceptance or rejection of the manuscript will depend on the completeness of your responses included in the next, final version of the manuscript. For this reason, and to save you from any frustrations in the end, I would strongly advise against returning an incomplete revision.

Summary and General Remarks

It is our pleasure to resubmit our revised manuscript on the effects of LSD1 inhibition on Merkel cell carcinoma (MCC). First, we would like to sincerely thank the editor and the referees for acknowledging the impact of the study and for their thoughtful points, which helped us to improve our study. In the revised manuscript, we have addressed all reviewer's comments and have

1. Strengthened the mechanistic analysis of the effect of LSD1 inhibition on MCC growth (*in vitro* and *in vivo*): We show that LSD1i reduces viability and inhibits cell growth in MCC. By performing various assays, including immunofluorescence and FACS-based staining for EdU/PI, AnnexinV/DAPI, cleaved caspase 3/7, PARP, Ki-67, and TUNEL, we found LSD1i treatment induces cell cycle arrest and cell death (new **Fig 3, point 2.1 and 2.2**).
2. Characterized the shift in cell fate induced by LSD1 inhibitor: We observed that LSD1i leads to an upregulation of neuronal differentiation markers in MCC and induces a gene expression signature resembling that of normal Merkel cells. By performing LSD1i drug wash-out experiments, we show that cells pretreated with LSD1i *in vitro* before injection into NSG mice have a strongly reduced tumor formation propensity *in vivo* and that the shift in cell fate is maintained in absence of LSD1i treatment (new **Fig 7, point 2.4**). These data indicate that optimized LSD1i dosing-regimes might lead to a therapeutic response while keeping systemic toxicity low.
3. Analyzed the effects of combination treatments: We performed combination treatments of drugs targeting the top-5 genetic dependencies identified from our epigenetic regulator analysis in MCC and of targetable members of LSD1-CoREST complex, namely HDAC1/2/3 (new **Fig EV2 and EV6**). We found that these drug combinations do not provide any gain compared to LSD1i single-agent treatments.
4. Discussed the Nature Cell Biology paper: We discussed the complementary study by Park *et al.* (Park *et al.*, 2020) in more detail and have also reanalyzed their CHIPseq datasets to validate the target genes identified in LSD1i-SLAMseq experiments (**point 2.6**).
5. Added additional HMG20B data: We have included additional data on HMG20B indicating that LSD1 inhibition in MCC disrupts the LSD1-CoREST complex by ablating the HMG20B protein. Additionally we show that all four HMG20B protein domains are necessary for MCC cell survival (new **Fig 6**).

We hope you are satisfied with our revisions and thank you again for contributing to the quality of this study.

Referee #1 (Remarks for Author):

In the present manuscript, the authors performed a pharmacological screen in Merkel cell carcinoma cells, targeting epigenetic regulators. They discovered that LSD1 is required for MCC growth in vitro and in vivo and that HMG20B (BRAF35), a subunit of the LSD1-CoREST complex, is also essential for MCC proliferation. LSD1 inhibition in MCC disrupts the LSD1-CoREST complex, directly activates TGF β signaling which induces the expression of regulators of the neuronal lineage, and activates a gene expression signature corresponding to normal Merkel cells. These results may offer a rationale for the use of LSD1 inhibitors against MCC.

The manuscript is well written and might, likely in short term, represent a real therapeutic strategy against Merkel cell carcinomas

Response: We thank the referee for acknowledging the relevance of our study and are eager to evaluate the efficacy of LSD1 inhibitors in clinical trials to improve the current treatment strategies for MCC patients.

Main Points

1.1: The authors analyze genome-wide RNAi screening data derived from the DepMap project, which includes genetic vulnerability maps for the MCC cell lines PeTa, MKL-1, and MKL-2. We examined the genes encoding the epigenetic regulators from their initial screen and found that LSD1 scores among the top 5 dependencies for MCC proliferation (Fig. 1e and Extended Data Fig. 1f). Would the finding of LSD1i applicability apply only to MCC cells or in general there would be a potential application for other neuroendocrine cancers? If neuroendocrine cancer cells different from MCC would be analysed, would LSD1 display a high score for dependencies for proliferation?

Response: We thank the reviewer for bringing up this interesting question. Neuroendocrine neoplasms originate primarily in the gastrointestinal tract (~63%) and the lung (~25%, mostly small-cell lung cancer, SCLC), but also in other organs (Oronsky *et al*, 2017). By interrogating dependencies for neuroendocrine tumors in the DepMap RNAi and CRISPR-KO dataset, we found that only SCLC tumors are sufficiently represented. LSD1 is indeed a strong dependency in some, but not all, SCLC cell lines (**Fig R1**), and could be a therapeutic option. The responsiveness of SCLC to LSD1i has been previously reported (Mohammad *et al*, 2015; Augert *et al*, 2019). Putting our findings in MCC in the broader context of studies investigating LSD1i treatment in other cancers, such as AMLs (Fang *et al*, 2019; Schenk *et al*, 2012; Somerville *et al*, 2016; Cai *et al*, 2020), SCLC (Mohammad *et al*, 2015; Augert *et al*, 2019), prostate cancer (Sehrawat *et al*, 2018), and cutaneous squamous cell carcinoma (Egolf *et al*, 2019), we believe that in certain cellular lineages, the neuroendocrine lineage among them, LSD1 is a gate-keeper of lineage plasticity, which can serve as a therapeutic entry point. We have highlighted this more extensively in the discussion.

Figure R1.1. Violin plot depicting the LSD1 dependency score in MCC compared to non-neuroendocrine skin cancer types, small-cell lung cancer (SCLC) and non-neuroendocrine lung cancer types. Red central line: median dependency score. Data obtained from the DepMap.

1.2: If possible, a part of the data should be recapitulated in primary MCC cells (if available), at least for response to LSDi and their anticancer potential *ex vivo* or in PDX.

Response: We agree that evaluating the effects of LSD1i in primary MCC cells or PDX models would be a nice addition. However, due to the rarity of MCC tumors, we did not have access to such model systems.

1.3: In figure 1e the dependency plot depicts the mean dependency of the 3 MCC cell lines PeTa, MKL-1, MKL-2 on the genes targeted by the compound library in Fig. 1a is shown. Would a combo scheme of LSD1i+BRD4i or other molecules targeting PRMT5, TAF1, WDR5 be more effective against MCC proliferation maintaining a similar specificity?

Response: We thank the referee for bringing up the concept of combination therapies and are very interested in combining LSD1i with other drug entities. As suggested by the reviewer, we combined the top-scoring epigenetic modifiers in our DepMap analysis for MCC, including the BRD4 inhibitor JQ1 (Filippakopoulos *et al*, 2010), the PRMT5 inhibitor GSK591 (Duncan *et al*, 2016), the TAF1 inhibitor BAY-299 (Bouché *et al*, 2017), and the WDR5 inhibitor OICR942 (Grebien *et al*, 2015) with the LSD1 inhibitor GSK-LSD1. First, we evaluated the sensitivity and specificity of all 5 drugs alone in two MCC cell lines (PeTa, MKL1) and human dermal fibroblasts (HDFB). The dose-response curves showed that similar to LSD1i, BRD4, and TAF1 inhibitors are effective in the nM range, whereas WDR5 and PRMT5 inhibitors are only effective at high μ M concentrations (new **Fig EV2A-D**). However, only LSD1i (GSK-LSD1) *selectively* inhibited growth in MCC cell lines; all of the other tested drugs also impaired the growth of HDFBs, indicating general toxicity (new **Fig EV2A-D**). Finally, we combined LSD1i (GSK-LSD1) with BRD4, PRMT5, TAF1, and WDR5 inhibitors to evaluate their synergistic effects; however, none of the tested drug combinations showed a significant synergistic effect (new **Fig EV2A-D**).

1.4: All the *in vivo* experiments have been performed in immune-compromised mice. If at all possible, the authors might include *in vivo* data in mice, which display a functional immune system. Alternatively, if technically this is not possible, the authors might comment on this issue in the discussion.

Response: Due to the lack of appropriate murine models recapitulating MCC pathogenesis in an immunocompetent background (Harms *et al*, 2018), we performed all our *in vivo* experiments with human MCC cell lines in immunocompromised NSG mice. While these models recapitulate important aspects of human MCC biology, they do not allow to assess the influence of LSD1i-induced tumor regression on the immune system. Given the growing evidence that a dedifferentiated state contributes to immune-evasion, it is indeed tempting to speculate that differentiation of MCC using LSD1 inhibitors could enhance responsiveness to checkpoint inhibitors. This is in line with a recent report indicating that LSD1 depletion enhances response to checkpoint inhibitors by activating a type 1 interferon response that stimulates a T cell response (Sheng *et al*, 2018). As suggested, we comment on this aspect in the discussion.

1.5: The authors identified core members, including the histone deacetylase HDAC2, and RCOR1, RCOR2, RCOR3, which serve as a scaffold for complex assembly, as well as non-canonical members, including GSE1, HMG20A, HMG20B (Fig. 3c and Extended Data Fig. 3a). Would a combo treatment targeting HDAC2 and LSD1 display any advantages against LSD1i as monotherapy?

Response: We thank the referee for this interesting question, especially because the combination of LSD1i with histone deacetylase (HDAC) inhibitors was shown to be superior over LSD1i alone in AML, glioblastoma and rhabdomyosarcoma in previous studies (Kalin *et al*, 2018; Anastas *et al*, 2019; Fiskus *et al*, 2014; Haydn *et al*, 2017). To assess synergistic effects of HDAC and LSD1i (GSK-LSD1) in MCC, we first assessed dose-response curves for the HDAC2 inhibitor, Santacruzamate A (Pavlik *et al*, 2013), and for an HDAC1/3 inhibitor, Entinostat (Saito *et al*, 1999), in two MCC cell lines (PeTa, MKL1) and HDFBs. MCC cells and HDFBs treated with Santacruzamate A alone showed both an IC50 of about \sim 80 μ M, indicating low sensitivity and no selectivity of MCC compared to HDFBs (new **Fig EV6I and J**). For cells treated with Entinostat, we observed IC50 levels of mid-nM concentrations

with an about 5-fold higher selectivity for MCC compared to HDFB cells (new **Fig EV2I**). After we determined the IC50 concentrations for these two drugs, we combined Santacruzamate A and Entinostat with GSK-LSD1 to assess their synergistic effect but could not identify a significant synergistic effect (new **Fig EV2I and J**).

1.6: The authors found that LSD1i treatment of MCC leads to dissociation of this complex, in line with studies in leukemias (Fig. 3d,ne). This evaluation is performed at 24h. Following, the authors to investigate the molecular programs associated with LSD1i mediated growth inhibition, performed RNAseq of DMSO- and GSK-LSD1-treated MCC cells after 6 days. The promoter motifs in the deregulated genes had Co-/Rest binding motif enriched in the promoters of genes that become up-regulated in MCCs upon inhibition of LSD1, indicating a direct regulation of these genes by the LSD1-CoREST complex (Fig. 4d). The authors thus suggest that the therapeutic effect of LSD1i treatment is mediated by the expression of neuronal genes and differentiating MCC cells into a neuronal lineage. Interestingly, HMG20B, was identified as a subunit of the LSD1-CoREST complex in MCC, required to maintain full repression of neuron-specific genes. Given that at 24 hours the authors see a change in the LSD1-CoREST co-repressor complex, which complex they think is binding to the responsive promoters at d6? And how the time d6 was chosen?

Response: We appreciate the reviewer's question, because it indicates that the presentation of our data might have been not clear enough. In the revised manuscript, we have streamlined the presentation and added additional data and hope that our data is now more clearly presented: In our co-IP MS analysis LSD1 of PeTa cells, we found that LSD1 interacts with the LSD1-CoREST complex members HDAC2, RCOR1-3 and the non-canonical proteins GSE1, HMG20A and HMG20B (**Fig 6A-C**). We found that 24h of LSD1i treatment is sufficient to disrupt the LSD1-CoREST complex (**Fig 6D and Fig EV6B**) and induce degradation of HMG20B (new **Fig 6F-H**). Interestingly, the degradation of HMG20B after 24h LSD1i treatment is maintained for at least 8 days *off drug* (new **Fig 7**). Since the LSD1-CoREST complex is durably disrupted, we investigated the transcriptional changes after 6 days of LSD1 inhibition which is also the readout endpoint of our proliferation assays (**Fig 1A and B**). In the promoter motif analysis, we found an enrichment of the Co-/REST promoter motif at de-regulated genes (**Fig 4G**), indicating that the sustained dissociation of the repressive LSD1-HMG20B-CoREST complex is important for de-repression of the LSD1 response genes.

1.7: By mapping the direct transcriptional response to LSD1i the authors identify TGF β signaling mediated activation of neuronal genes, suggesting that transcriptional activation of TGF β signaling in LSD1i-treated MCCs phenocopies receptor-ligand based TGF β activation and induces neuronal differentiation. A plethora of small molecules targeting inhibition of TGF β pathway, inhibition of BMP receptors, SMADs etc are commercially available and their use (with potential interference or synergy depending on the molecule chosen) in these settings might further prove the authors hypothesis even potentially suggesting the role of the different partners as well as additional therapeutic options.

Response: We genuinely thank the referee for this excellent suggestion, which allowed us to further characterize the induction of TGF β and its role in neuronal differentiation upon LSD1i. To this end, we treated PeTa cells with several small molecules (SB431542, TGF β , Activin, BMP-2, BMP-4, Noggin, Galunisertib, LDN-193189) acting on receptors or downstream mediators of the TGF β pathway, alone or in combination with LSD1i (**Fig R1.7A**).

The activation of the canonical TGF β signaling via TGF β or Activin did not induce neuron differentiation-specific genes such as ID1, ID3, SOX3 or NEUROD1 (**Fig R1.7B and C**). The pharmacological inhibition of TGFBR1 with Galunisertib and the inhibition of ACVR1B/TGFBR1/ACVR1C with SB431542 did not prevent the LSD1i-mediated induction of those neuronal differentiation genes (**Fig R1.7D and E**). Collectively, these data indicate that canonical TGF β signaling via SMAD2/3 phosphorylation is not involved in LSD1i-mediated induction of neuronal differentiation genes in MCC.

The combination of LSD1i with Noggin, a negative regulator of the non-canonical (BMP arm of) TGF β signaling, ablated expression of ID3, but not of neuronal markers SOX3, NEUROD1, ANK3 and SYT4 (**Fig R1.7G**). The combination of LSD1i with inhibition of ACVR1/BMPR1A with LDN-193189, reduced phosphorylation of

SMAD1/5/9 and blocked the expression of ID1 and ID3 in the presence of LSD1i, but did not alter the induction of neuronal markers (**Fig R1.7G and H**). In summary, we conclude that LSD1 inhibition strongly induces the non-canonical BMP arm of TGF β signaling, but it is not a driver of the neuronal gene expression induced by LSD1i treatment and have revised the figures and text accordingly.

Figure R1.7. **A.** Schematics of the TGF β signaling pathway with compounds and targets tested herein. **B-G.** RT-qPCR of TGF β pathway components and neuromodulators upon indicated treatment. Data are represented relative to the housekeeping gene HPRT1 and

normalized to the negative control, DMSO. n = 4 technical replicates. H. Western blot confirmation of SMAD1/5/9 phosphorylation upon LDN193189 treatment. H3 serves as loading control.

Minor Points

1.8: The authors might enlarge the LSD1i tested in MCC cell lines to a broader panel including some of them such as TCP, IMG-7289, INCB059872, CC-90011, and ORY-2001

Response: To elucidate important biological aspects of LSD1 inhibition in MCC, we used two independent LSD1 inhibitors (GSK-LSD1, ORY-1001), which fall both in the class of irreversible LSD1 inhibitors. Both inhibitors are structural derivatives of tranylcypromine (TCP), inhibit MCC proliferation at low nM-IC50 values, and show selectivity in MCC compared to HDFB (**Fig 1A-C and Fig EV1A**). In our manuscript, we confirmed many effects with both pharmacological compounds (e.g., dose-response experiments, biochemical assays and *in vivo* experiments). We think that 2 LSD1i are sufficient to highlight the underlying biology of LSD1 inhibition in MCC. However, our ultimate goal is to make a clinical impact and to improve the therapy for MCC patients. In this regard, we completely agree with the reviewer that testing additional compounds is essential. To prepare the ground for an investigator-initiated clinical trial, we are currently in the process of selecting the most promising LSD1i: ORY-1001, GSK-LSD1 (which is closely related to the clinical compound GSK-2879552) and other, irreversible LSD1i (e.g., IMG-7289, INCB059872) were/are under clinical evaluation for cancer treatment. However, irreversible LSD1i are often considered as having an unfavorable efficacy/safety profile and the results of some phase I clinical trials have been sobering (e.g., GSK-2879552: [clinicaltrials.gov: NCT02034123](https://clinicaltrials.gov/ct2/show/study/NCT02034123), [NCT02177812](https://clinicaltrials.gov/ct2/show/study/NCT02177812), [NCT02929498](https://clinicaltrials.gov/ct2/show/study/NCT02929498)). Other compounds are not the first choice for MCC due to their poor and unspecific LSD1 inhibition: TCP (tranylcypromine) is mainly a monoamine oxidase (MAO) inhibitor, which only poorly inhibits LSD1, (Fang *et al*, 2019), and is mainly used to treat neurological disorders such as depression ([clinicaltrials.gov: NCT02717884](https://clinicaltrials.gov/ct2/show/study/NCT02717884)). Similarly, ORY-2001, structurally related to ORY-1001 (Maes *et al*, 2016, 2017), is a dual inhibitor of MOAB and LSD1 and under clinical investigation for patients with mild to moderate Alzheimer's disease ([clinicaltrials.gov: NCT03867253](https://clinicaltrials.gov/ct2/show/study/NCT03867253)).

More recently, structures of potent, selective, and reversible LSD1i are emerging (Romussi *et al*, 2020), which might offer a better efficacy/safety profile and would be more promising to initiate clinical trials for MCC patients. CC-90011 is the first reversible LSD1i, which is investigated in several clinical trials for patients with advanced unresectable neuroendocrine tumors, including SCLC and other neuroendocrine carcinomas, Non-Hodgkin lymphoma and non-small cell lung carcinoma (NSCLC) ([clinicaltrials.gov: NCT02875223](https://clinicaltrials.gov/ct2/show/study/NCT02875223), [NCT04350463](https://clinicaltrials.gov/ct2/show/study/NCT04350463), [NCT03850067](https://clinicaltrials.gov/ct2/show/study/NCT03850067)). Thus, to pave the way for a phase I clinical trial, we will focus on the newer potent, selective, and reversible LSD1i. However, in-depth mechanistic studies that provide guidance on the design of dosing schedules will be necessary. Experiments in humanized mouse models to investigate whether LSD1i in MCC not only induces neuronal differentiation of MCC, but simultaneously improves the response to immune checkpoint inhibitors, as suggested by a recent study (Sheng *et al*, 2018) will be interesting. However, these preclinical experiments will take several months and exceed the timeline and scope of this manuscript, focusing on the underlying biology of LSD1 inhibition in MCC. Nevertheless, we want to thank the referee for this interesting suggestion.

Referee #2 (Remarks for Author):

Review manuscript EMBO Molecular Medicine- EMM-2020-12525

"LSD1 inhibitors induce neuronal differentiation of Merkel cell carcinoma"

The manuscript from Dr. Obenaus and colleagues entitled "LSD1 inhibitors induce neuronal differentiation of Merkel cell carcinoma" describes a vulnerability of Merkel Cell Carcinoma (MCC) cells to pharmacological treatment with inhibitors of the histone demethylase LSD1, unraveling part of the downstream events leading to differentiation of the MCC cells.

Specifically, the authors perform a pharmacological screen in MCC cells discovering a high sensitivity for LSD1 inhibitors treatment, and further consolidate this finding with knockdown experiments and data mining. The conclusion that MCC cells are dependent from LSD1 activity for their *in vitro* proliferation is then translated *in vivo*, showing that LSD1 inhibition affects tumor maintenance and tumor establishment in MCC xenograft models. The authors proceed to identify via IP-Mass Spectrometry interactors of LSD1, suggesting a prominent function of RCOR1 and of the non-canonical BRAF-histone deacetylase complex member HMG20B. Finally, the authors investigate via gene expression profiling and cell biology techniques the events downstream of LSD1 inhibition, identifying a function of LSD1 in inhibiting neuronal differentiation genes in MCC cells. This signature is overlapping with the expression profile of normal Merkel cells, and overall the data hint at the fact that LSD1 inhibition causes the de-repression of neuronal differentiation genes, thereby triggering a shift in cell fate that induces MCC to become normal Merkel cells. Finally, by profiling the newly transcribed RNAs with SLAMseq at different time points after LSD1 inhibition, the authors identify members downstream components of BMP signaling as bona fide effectors of the neuronal differentiation induction following LSD1 inhibition.

The work is well presented, technically sound and well controlled. The lack of treatment options for Merkel Cell Carcinoma makes the findings very interesting in terms of potential clinical applications, and the link between the LSD1 inhibitors treatment and the induction of physiological differentiation is very relevant. Part of the findings of this paper is overlapping with the recently published paper by Park and colleagues ("Merkel cell polyomavirus activates LSD1-mediated blockade of non-canonical BAF to regulate transformation and tumorigenesis", *Nature Cell Biology* 2020). Considering that the paper was published concomitantly of the submission, the non-scooping policy of EMBO related journal should apply to this case.

Following are some general recommendations that would improve the quality of the manuscript:

Response: We thank the reviewer for the excellent suggestions, the positive words, and for mentioning the applicability of the non-scooping policy of EMBO related journals.

2.1: The authors claim that LSD1 inhibitor treatment induces arrest of MCC cells growth *in vitro* and of tumor growth *in vivo*. The conclusion is that this is not due to cell death, but rather a decrease in the speed of the cell cycle due to the induction of a differentiation program.

For the *in vitro* experiments, this conclusion is supported (in Fig 4 and Extended Data Fig 4) by morphological changes of the cells, changes in gene expression, caspase dependent apoptosis assays (PARP cleavage and Caspase cleavage) and KI-67 expression. Since this is a major point of the paper, the authors should rule out the possibility of other forms of caspase independent cell death (for instance, by performing TUNEL assay or AnnexinV staining) to prove that the decrease in cell number is really not due to any contribution of cell death. Also, concerning the cell cycle analysis, it would be advisable to extend it by performing BrdU/EdU labeling experiments, in order to describe in detail if the cell cycle is slowed down or arrested. Finally, concerning the induction of differentiation, I would like to have a validation of the expression of some Merkel cells markers by immunofluorescence, to prove that LSD1 inhibition really results in a change in cell identity.

Response: We sincerely thank the reviewer for these excellent suggestions that allowed us to dig deeper into the effects of LSD1i in MCC and extend our initial findings. While LSD1i did not induce PARP or caspase-cleavage *in vitro* (new **Fig 3D and E**), as suggested, we assessed other forms of caspase-independent cell death. We identified time-dependent mitochondrial depolarization over 6 days of LSD1i [100 nM GSK-LSD1] using tetramethylrhodamine (TMRE), an early marker of cell death (new **Fig 3F and G**). We performed Annexin V/DAPI

staining of cells treated with LSD1i for 6 days *in vitro*, which revealed an increase from ~8 to ~50 % of Annexin V-positive cells (new **Fig 3H and I**). Immunofluorescence stainings for TUNEL and Ki-67 on cells treated *in vitro* with LSD1i, showed a 2-fold increased TUNEL staining, indicating increased DNA fragmentation induced during cell death and a reduction from 33% to 10 % in proliferation (Ki-67) (**Fig 3A and J**). To refine the effect on cell cycle progression, we performed EdU/PI labeling experiments and treated PeTa cells for 3 and 6 days with 100 nM GSK-LSD1 or DMSO. In line with the slow doubling time of PeTa cells *in vitro*, we found that ~8% of untreated (DMSO) cells are actively undergoing DNA synthesis in S-phase, but this was reduced upon LSD1i treatment, with only ~4% of cells in S-phase. Moreover, after 3 days of LSD1i treatment, we observed an initial increase of ~7% of the G0/1 population, whereas after 6 days of LSD1i treatment, this difference in the G0/1 population was diminished (simultaneously we found an increase in cell death), indicating a G0/1 cell cycle arrest subsequently leading to cell death (new **Fig 3B and C, Fig EV 4A**). In line with these results, our transcriptome dataset showed a strong downregulation of genes involved in cell cycle control and progression, after 6 days of LSD1i treatment (**Fig 4D and E, Fig EV 5C**). Altogether, these data indicate that the LSD1i-induced expression of fate-determining transcription factors (**Fig 5B**) and differentiation of MCC cells into a neuronal lineage enriched in features of normal Merkel cells (**Fig 5F-H**) leads to an inhibition in cell growth and caspase-independent cell death *in vitro*. These new data are now prominently highlighted in the manuscript and reflected in the title.

Regarding the validation of expression of Merkel cell markers by immunofluorescence, we are facing the challenge that all established normal Merkel cell markers, such as Cytokeratin 20, CD56, chromogranin and synaptophysin, are already expressed in Merkel cell carcinoma, precluding the possibility to make a qualitative statement about differentiation using immunofluorescence.

2.2: The same concepts apply to the *in vivo* experiments described in Fig 2 and in Extended Data Fig. 2. In the plot in Fig. 2B, it seems that some tumors achieve a "steady state" size, some others have a slight increase, others a small decrease. In Ext. Data Fig.2A, which focuses on the early time points after the treatment, the impression is that overall tumors are still slowly growing for some time. It would be interesting to perform a histological analysis of the tumors at different time points (e.g. few days after the start of the treatment, at mid-term and at a later time point) to verify what is happening in the tumor mass. Are the cells switching completely identity and undergoing terminal differentiation? Is a part of the tumor undergoing cell death? Is there a significant remodeling of the microenvironment that could explain the slowdown in tumor growth? The effect on tumor maintenance and establishment is one of the most intriguing findings of this paper, and should definitely be analyzed in a more detailed manner.

Response: We thank the reviewer for bringing up the interesting aspect of how LSD1i changes the histological and immunohistological appearance. As suggested, we performed H&E staining and immunofluorescence of MCC tumors treated with LSD1i or vehicle. After the tumors reached a size of $\geq 50\text{mm}^3$, we started treatment with LSD1i and explanted the tumors after 1 day ("D1"), 10 days ("D10"), or at the experimental endpoint ("Endpoint"). The MCC tumors in vehicle-treated mice showed relatively uniform, small round to oval cells with round nuclei and scant cytoplasm. LSD1 treated tumors showed slightly larger and elongated cells with irregular shaped nuclei and ill-defined cell borders, but no complete switch of cell identity (new **Fig 3M**). At all time points, we observed a ~50% decrease in cell proliferation (Ki-67 staining, new **Fig 3K**) and a ~5-10x increase in apoptosis reflected in TUNEL staining (new **Fig 3L**). In contrast to our observation *in vitro*, we also identified an up to ~4-fold increase in cleaved caspase-3 staining (new **Fig EV4C**). In summary, LSD1i-induced gene expression changes translated into morphological changes in MCC cells and increased cell death *in vivo*.

2.3: - To make the work more complete, the authors should assess the relative importance of the genes that are induced after LSD1 inhibition, for example by trying to mimic the differentiation induction upon genetic manipulation in the absence of LSD1 inhibition, e.g. by inducing ectopic expression of some of the factors that are implicated in Merkel cells normal differentiation (The alternative approach would be showing the absence

of differentiation induction after LSD1 inhibitors treatment upon knockdown of one or more of the genes implicated in the neuronal differentiation program.

Response: We thank the referee for proposing this experiment. We overexpressed the *bona fide* LSD1 target NEUROD1 (4-fold upregulation upon LSD1i treatment, new **Fig 5B**), which has been described as a regulator of ATOH1 - a master transcription factor in normal Merkel cell development - and a key regulator of neuronal differentiation programs (Mulvaney & Dabdoub, 2012; Boutin *et al*, 2010) and subsequently performed RT-qPCR analysis on LSD1i-response genes in MCC (**Fig R2.3**). We find that overexpression of NEUROD1 alone is sufficient to partially recapitulate the induction of differentiation in part by upregulating SOX3 (3-fold), ANK3 (2-fold) as well as ID1 (2-fold), compared to EV control. However, we would like to note that the expression changes were more modest in comparison to inhibition of LSD1 and expression levels of SYT4 were not changed upon NEUROD1 overexpression. These data indicate that the expression of a single factor such as NEUROD1 is able to drive part of the neuronal differentiation program, but it is not sufficient to fully recapitulate the transcriptional program induced by LSD1i treatment. It is likely that the induced shift in cell fate is not caused by the specific expression of a single cell-fate defining factor but by the concerted deregulation of multiple factors that together orchestrate the induction of a normal Merkel cell-like transcriptional program and induce cell death in MCC.

Figure R2.3. RT-qPCR quantification of neuronal genes after ectopic expression of NEUROD1 or empty vector (EV) control. Data are relative to the housekeeping gene HPRT1 and normalized to the respective DMSO control. n = 4 technical replicates. Bar graphs represent mean \pm SD.

2.4: - Also, it would be interesting to see how stable the shift in cell fate is. For instance, are MCC cells pre-treated with LSD1 inhibitors able to form tumors (so not at day1 after grafting, but at day -2 or -3)? If not (which would be my expectation), how long it takes for them to revert to a pro-tumorigenic phenotype, if this is at all possible, and what are the downstream events that are involved (e.g. it's merely a reactivation of LSD1?)

Response

We thank the reviewer for this very intriguing question that allowed us to further expand our manuscript regarding the durability of the LSD1i-mediated shift in cell fate and allowing us to highlight the potency of LSD1i treatment in MCC. To assess if the LSD1i treatment induces a stable shift in cell fate, we treated PeTa cells for 24h with 100 nM GSK-LSD1 or DMSO, washed the drug out and harvested cells for 8 days (new **Fig 7A**).

We found by RT-qPCR that a 24h LSD1i-pulse is sufficient to induce sustained transcriptional upregulation of direct LSD1 targets of the neuronal lineage, such as ANK3, SOX3, SYT4, or NEUROD1 (new **Fig 7B**). Additionally, we found that HMG20B protein levels were strongly decreasing over 8 days, suggesting a maintained disruption of the LSD1-CoREST complex in MCC (new **Fig 7D**). For other LSD1-CoREST complex members, we did not observe changes in protein levels (new **Fig 7D**). We also found that the induction of TGF β pathway members, indicated by elevated levels of phospho-SMAD1/5/9 and ID1, was maintained during the drug-washout period (new **Fig 7C**). To uncover if maintained activation of those programs is sufficient to induce a reduction in cell proliferation, we performed EdU/PI and Annexin V/DAPI stainings. Similar to our previous results obtained in the presence of LSD1i (new **Fig 3**), during the drug-washout period, we observed a G0/1 cell cycle arrest (new **Fig 7E and F**) together with strong induction of cell death (new **Fig 7C and G**). Finally, we probed whether the shift in cell fate

is also stable *in vivo*. We pre-treated PeTa cells, either for 1, 3, or 6 days with 100 nM GSK-LSD1 and found that pretreated cells have a strongly reduced tumor formation propensity in the absence of additional drug (new **Fig 7H and I**). Altogether these data indicate that pharmacological LSD1i induces a stable shift in cell fate in MCC, driven by the disassembly of the LSD1-CoREST complex and sustained activation of master regulators of the neuronal lineage.

2.5: In connection to this, is it conceivable that following LSD1 inhibitor treatment other epigenetic regulation complexes could take over the repression of pro-differentiation genes? Did the authors observe any upregulation of other demethylases that could suggest the acquisition of resistance mechanisms?

Response: We investigated the transcriptional changes after 6 days of LSD1i treatment and found that multiple chromatin modifier genes (GO:0016570) were differentially expressed; however, we could not identify a gene - that to the best of our knowledge - can compensate for LSD1 loss (**Fig R2.5**). Second, we analyzed the LSD1 protein structure and found that it harbors a coiled-coil protein domain, that allows it to bind to the CoREST complex (Burg *et al*, 2015; Yang *et al*, 2006). Interestingly, closely related proteins such as the demethylase KDM1B, but also PAOX, SMOX, IL4I1, MAOA and MAOB, do not have this domain and can therefore not take LSD1s function within the LSD1-CoREST complex. Finally, we treated PeTa and MKL1 cells for 12 weeks with a sub-IC50 concentration of 1 nM GSK-LSD1 but could not observe the development of resistance.

Figure R2.5. Volcano plot of members of the GO term histone modifications transcriptionally deregulated after 6 days of 100 nM GSK-LSD1 treatment.

2.6: - My final remark is about the paper by Park *et al*. Obenauf and colleagues refer to this paper very briefly stating that "Interestingly, it was recently reported that the small T antigen of the Merkel cell polyomavirus establishes a dependency on LSD1 in MCC". The actual points of cross-talk between the work done by the two groups are quite numerous, and it would be a good idea to refer to the other paper a bit more extensively, even possibly by making use of the data when useful (e.g. of the CHIP-seq data).

Response: We fully agree with the reviewer and apologize for not appropriately putting our work in the context of the Park *et al*. paper. In our revised manuscript, we discuss the Park *et al*. paper in more detail and use the LSD1-RCOR2 CHIP-seq dataset to validate our SLAM-seq target genes (**Fig R2.6**). Interestingly, we find that while most of the direct LSD1 target genes identified in our metabolic labeling approach (SLAM-seq) show binding of LSD1 and RCOR2 within their promoter region, the high number of additional LSD1-RCOR2 binding sites without alteration of gene expression, indicates that promoter binding alone does not necessitate gene regulation by the LSD1-CoREST complex.

Figure R2.6. Upset plot representing the overlap of the differentially regulated genes identified in the SLAMseq experiment with the binding sites of RCOR2 and LSD1 identified in ChIPseq experiments by *Park et al.* (Park et al, 2020).

Referee #3 (Comments on Novelty/Model System for Author):

It is a very thorough study with many different methods applied. The novelty is, however, limited since a recent publication demonstrated that Merkel cell polyomavirus encoded small T Antigen activates LSD-1 expression, and that LSD1 inhibition has an effect on MCC growth in vitro and in vivo.

Response: We thank the reviewer for describing our manuscript as very thorough. Regarding the novelty, we think that it is encouraging that DeCaprio's and our group come independently to very similar conclusions despite major differences, such as the studies entry points (we performed a small-molecule screen to identify vulnerabilities, while DeCaprio's lab comes from an etiological angle), the experiments and the methods. Moreover, we would like to emphasize that our work also highlights the concept of targeting cell fate regulators in solid cancers. Finally, we would like to refer to EMBO's 'scooping protection' policy.

Referee #3 (Remarks for Author):

In the manuscript of Leiendecker et al the authors performed a pharmacological screen in Merkel cell carcinoma (MCC) cells targeting epigenetic regulators. This analysis revealed that lysine-specific histone demethylase 1A (LSD1/KDM1A) is required for MCC growth. They further demonstrate that LSD1 inhibition triggers the TGF β signaling pathway resulting in the expression of key regulators of the neuronal lineage and a Merkel cell like gene signature.

As the authors pointed out, currently only immune checkpoint inhibitors are approved for therapy of metastatic patients, which sometimes have limited efficiency. Thus, there is still a medical need for efficient therapies for MCC. Since in virus-positive tumors no prominent mutations are present, targeted therapies directed against epigenetic regulators appear as a promising alternative. Therefore, the object of the study is absolutely justified. The authors present a well-written, thorough and experimental sound study. Their results are in line with a recent publication by Park et al (Nat Cell Biol., 2020) in which the authors demonstrate that Merkel cell polyomavirus (MCPyV) encoded small T antigen (sT) activates the expression of LSD1 rendering MCC cells sensitive to LSD1 inhibition.

Points, which should be addressed by the authors:

Response: We thank the reviewer for her/his kind words describing our objective as "absolutely justified", and the study as "well-written, thorough and experimental". We agree that our results are in line with the recent publication by Park et al (Park *et al*, 2020). Although this independent validation of the underlying biological basis of LSD1 inhibition in MCC came as a surprise, in times of poor reproducibility in biomedical research (Baker, 2016), it will increase the chances to start phase 1 clinical trials to evaluate the efficacy of LSD1 inhibitors in MCC and might ultimate help to improve the therapy for MCC patients.

3.1: For their analyses they only used MCPyV-positive MCC cells. The effect on virus-negative cells has not been tested. Therefore, the authors should make it clear that their observation only applies to the virus-positive subgroup.

Response: We apologize for the imprecise presentation of our findings that LSD1 is a dependency in virus-positive MCC. We changed the wording in the manuscript to highlight that the herein described LSD1-dependency only applies to virus-positive MCC tumors. However, we would like to point out that to our best knowledge no representative virus-negative MCC model is currently available. The 3 publicly available, virus-negative MCC lines (UISO, MCC13, and MCC26) are controversial (Park *et al*, 2020; Daily *et al*, 2015). All virus-negative MCC models show a gene expression profile, growth pattern and immunoprofile that is different from fresh frozen MCC tumors, whereas that of the virus-positive MCC cell lines WaGa and MKL-1 is similar to fresh frozen MCC (Daily *et al*, 2015). Due to lack of well established and representative virus-negative MCC models, we currently do not know if LSD1 is an exploitable vulnerability in virus-negative MCC.

3.2: The presentation of the dependency score as violin and box plots for groups with only a few data points (three in the case of MCC) seems inappropriate.

Response: We thank the referee for pointing out the inappropriate presentation of our dependency score plots. We depict now all individual data points, median, and a violin plot visualizing the underlying data distribution in the revised manuscript (**Fig 1F, Fig 6E and EV6D-E**).

3.3: The authors couldn't detect apoptotic death upon LSD1 inhibitor treatment. Accordingly, the mechanism how LSD1 inhibition affects MCC cell growth is not clear. Probably a mixture between cell death and cell cycle arrest. To scrutinize the effect, the authors should perform cell counting and cell cycle analysis with Edu/Brdu staining.

Response: We thank the reviewer for suggesting to delineate the effects that lead to cell death in MCC upon LSD1i. To assess the effect of LSD1i on cell cycle progression, we performed EdU/PI labeling and treated PeTa cells for 3 and 6 days with 100 nM GSK-LSD1 or DMSO. In line with the slow doubling time of PeTa cells *in vitro*, we found that ~8% of untreated (DMSO) cells are actively undergoing DNA synthesis in S-phase, with a reduction to only ~4% of cells in S-phase upon LSD1i (new **Fig 3B and C**). After 3 days of LSD1i treatment, we observed an increase of ~7% of the G0/1 population, whereas after 6 days, this difference in the G0/1 population was diminished (simultaneously we found a strong increase in cell death), indicating a G0/1 cell cycle arrest subsequently leading to cell death (new **Fig 3B and C, Fig EV 4A**).

While LSD1i treatment did not induce PARP or caspase-cleavage *in vitro*, (new **Fig 3D and E**) we assessed other forms of caspase independent cell death. Interestingly, we identified time-dependent mitochondrial depolarization over 6 days of LSD1i [100 nM GSK-LSD1] using tetramethylrhodamine (TMRE), an early marker of cell death (new **Fig 3F and G**). Moreover, we performed Annexin V/DAPI staining of cells treated with LSD1i for 6 days *in vitro*, which revealed an increase from ~8 to ~50 % of Annexin V-positive cells (new **Fig 3H and I**). Immunofluorescence stainings for TUNEL and Ki-67 on cells treated *in vitro* with LSD1i, showed a 2-fold increased TUNEL staining, indicating an increased DNA fragmentation induced during cell death and a reduction from 33% to 10 % in proliferation (Ki-67) (new **Fig 3A and J**). Altogether, these data indicate that the LSD1i treatment leads to a cell cycle arrest and caspase-independent cell death *in vitro*. Our observations are in line with a previous report (Houben *et al*, 2010), that showed that inhibition of the T antigens induces cell cycle arrest with subsequent caspase-independent cell death. These data suggest that T antigen-mediated transformation relies on LSD1 to lock MCC cells in a highly proliferative stem-like state.

3.4: According to figure 1b even at the highest doses of inhibitor the viability stays at about 25%. What happens in long time culture experiments? Do they become drug resistant?

Response: We thank the reviewer for this question. As it is our ultimate goal to make a clinical impact and to improve the therapy for MCC patients by treatment with LSD1i alone or in combination with immunotherapy, we are very interested in the development of resistance to LSD1i in MCC. To this end, we treated the MCC cell lines PeTa and MKL1 for 12 weeks with a sub-IC50 concentration of 1 nM GSK-LSD1, however, were not able to observe the development of resistance.

References

- Anastas JN, Zee BM, Kalin JH, Kim M, Guo R, Alexandrescu S, Blanco MA, Giera S, Gillespie SM, Das J, Wu M, Nocco S, Bonal DM, Nguyen Q-D, Suva ML, Bernstein BE, Alani R, Golub TR, Cole PA, Filbin MG, et al (2019) Re-programing Chromatin with a Bifunctional LSD1/HDAC Inhibitor Induces Therapeutic Differentiation in DIPG. *Cancer Cell* Available at: <http://dx.doi.org/10.1016/j.ccell.2019.09.005>
- Augert A, Eastwood E, Ibrahim AH, Wu N, Grunblatt E, Basom R, Liggitt D, Eaton KD, Martins R, Poirier JT, Rudin CM, Milletti F, Cheng W-Y, Mack F & MacPherson D (2019) Targeting NOTCH activation in small cell lung cancer through LSD1 inhibition. *Sci. Signal.* **12**: Available at: <http://dx.doi.org/10.1126/scisignal.aau2922>
- Baker M (2016) 1,500 scientists lift the lid on reproducibility. *Nature* **533**: 452–454
- Bouché L, Christ CD, Siegel S, Fernández-Montalván AE, Holton SJ, Fedorov O, Ter Laak A, Sugawara T, Stöckigt D, Tallant C, Bennett J, Monteiro O, Díaz-Sáez L, Siejka P, Meier J, Pütter V, Weiske J, Müller S, Huber KVM, Hartung IV, et al (2017) Benzoisoquinolinediones as Potent and Selective Inhibitors of BRPF2 and TAF1/TAF1L Bromodomains. *J. Med. Chem.* **60**: 4002–4022
- Boutin C, Hardt O, de Chevigny A, Coré N, Goebbels S, Seidenfaden R, Bosio A & Cremer H (2010) NeuroD1 induces terminal neuronal differentiation in olfactory neurogenesis. *Proc. Natl. Acad. Sci. U. S. A.* **107**: 1201–1206
- Burg JM, Makhoul AT, Pemble CW 4th, Link JE, Heller FJ & McCafferty DG (2015) A rationally-designed chimeric KDM1A/KDM1B histone demethylase tower domain deletion mutant retaining enzymatic activity. *FEBS Lett.* **589**: 2340–2346
- Cai SF, Chu SH, Goldberg AD, Parvin S, Koche RP, Glass JL, Stein EM, Tallman MS, Sen F, Famulare CA, Cusan M, Huang C-H, Chen C-W, Zou L, Cordner KB, DelGaudio NL, Durani V, Kini M, Rex M, Tian HS, et al (2020) Leukemia cell of origin influences apoptotic priming and sensitivity to LSD1 inhibition. *Cancer Discov.* Available at: <http://dx.doi.org/10.1158/2159-8290.CD-19-1469>
- Daily K, Coxon A, Williams JS, Lee C-CR, Coit DG, Busam KJ & Brownell I (2015) Assessment of cancer cell line representativeness using microarrays for Merkel cell carcinoma. *J. Invest. Dermatol.* **135**: 1138–1146
- Duncan KW, Rioux N, Boriack-Sjodin PA, Munchhof MJ, Reiter LA, Majer CR, Jin L, Johnston LD, Chan-Penebre E, Kuplast KG, Porter Scott M, Pollock RM, Waters NJ, Smith JJ, Moyer MP, Copeland RA & Chesworth R (2016) Structure and Property Guided Design in the Identification of PRMT5 Tool Compound EPZ015666. *ACS Med. Chem. Lett.* **7**: 162–166
- Egolf S, Aubert Y, Doepner M, Anderson A, Maldonado-Lopez A, Pacella G, Lee J, Ko EK, Zou J, Lan Y, Simpson CL, Ridky T & Capell BC (2019) LSD1 Inhibition Promotes Epithelial Differentiation through Derepression of Fate-Determining Transcription Factors. *Cell Rep.* **28**: 1981–1992.e7
- Fang Y, Liao G & Yu B (2019) LSD1/KDM1A inhibitors in clinical trials: advances and prospects. *J. Hematol. Oncol.* **12**: 129
- Filippakopoulos P, Qi J, Picaud S, Shen Y, Smith WB, Fedorov O, Morse EM, Keates T, Hickman TT, Felletar I, Philpott M, Munro S, McKeown MR, Wang Y, Christie AL, West N, Cameron MJ, Schwartz B, Heightman TD, La Thangue N, et al (2010) Selective inhibition of BET bromodomains. *Nature* **468**: 1067–1073
- Fiskus W, Sharma S, Shah B, Portier BP, Devaraj SGT, Liu K, Iyer SP, Bearss D & Bhalla KN (2014) Highly effective combination of LSD1 (KDM1A) antagonist and pan-histone deacetylase inhibitor against human AML cells. *Leukemia* **28**: 2155–2164

- Grebien F, Vedadi M, Getlik M, Giambruno R, Grover A, Avellino R, Skucha A, Vittori S, Kuznetsova E, Smil D, Baryte-Lovejoy D, Li F, Poda G, Schapira M, Wu H, Dong A, Senisterra G, Stukalov A, Huber KVM, Schönegger A, et al (2015) Pharmacological targeting of the Wdr5-MLL interaction in C/EBP α N-terminal leukemia. *Nat. Chem. Biol.* **11**: 571–578
- Harms PW, Harms KL, Moore PS, DeCaprio JA, Nghiem P, Wong MKK, Brownell I & International Workshop on Merkel Cell Carcinoma Research (IWMCC) Working Group (2018) The biology and treatment of Merkel cell carcinoma: current understanding and research priorities. *Nat. Rev. Clin. Oncol.* **15**: 763–776
- Haydn T, Metzger E, Schuele R & Fulda S (2017) Concomitant epigenetic targeting of LSD1 and HDAC synergistically induces mitochondrial apoptosis in rhabdomyosarcoma cells. *Cell Death Dis.* **8**: e2879
- Houben R, Shuda M, Weinkam R, Schrama D, Feng H, Chang Y, Moore PS & Becker JC (2010) Merkel cell polyomavirus-infected Merkel cell carcinoma cells require expression of viral T antigens. *J. Virol.* **84**: 7064–7072
- Kalin JH, Wu M, Gomez AV, Song Y, Das J, Hayward D, Adejola N, Wu M, Panova I, Chung HJ, Kim E, Roberts HJ, Roberts JM, Prusevich P, Jeliakov JR, Roy Burman SS, Fairall L, Milano C, Eroglu A, Proby CM, et al (2018) Targeting the CoREST complex with dual histone deacetylase and demethylase inhibitors. *Nat. Commun.* **9**: 53
- Maes T, Mascaró C, Rotllant D, Cavalcanti F, Carceller E, Ortega A, Molinero C & Buesa C (2016) ORY-2001: An epigenetic drug for the treatment of cognition defects in alzheimer's disease and other neurodegenerative disorders. *Alzheimers. Dement.* **12**: P1192
- Maes T, Molinero C, Antonijoan RM, Ferrero-Cafiero JM, Martínez-Colomer J, Mascaró C & Arevalo MI (2017) First-in-human phase I results show safety, tolerability and brain penetrance of ORY-2001, an epigenetic drug targeting LSD1 and MAO-B. *Alzheimers. Dement.* **13**: P1573–P1574
- Mohammad HP, Smitheman KN, Kamat CD, Soong D, Federowicz KE, Van Aller GS, Schneck JL, Carson JD, Liu Y, Butticello M, Bonnette WG, Gorman SA, Degenhardt Y, Bai Y, McCabe MT, Pappalardi MB, Kaspavec J, Tian X, McNulty KC, Rouse M, et al (2015) A DNA Hypomethylation Signature Predicts Antitumor Activity of LSD1 Inhibitors in SCLC. *Cancer Cell* **28**: 57–69
- Mulvaney J & Dabdoub A (2012) Atoh1, an essential transcription factor in neurogenesis and intestinal and inner ear development: function, regulation, and context dependency. *J. Assoc. Res. Otolaryngol.* **13**: 281–293
- Oronsky B, Ma PC, Morgensztern D & Carter CA (2017) Nothing But NET: A Review of Neuroendocrine Tumors and Carcinomas. *Neoplasia* **19**: 991–1002
- Park DE, Cheng J, McGrath JP, Lim MY, Cushman C, Swanson SK, Tillgren ML, Paulo JA, Gokhale PC, Florens L, Washburn MP, Trojer P & DeCaprio JA (2020) Merkel cell polyomavirus activates LSD1-mediated blockade of non-canonical BAF to regulate transformation and tumorigenesis. *Nat. Cell Biol.* Available at: <http://dx.doi.org/10.1038/s41556-020-0503-2>
- Pavlik CM, Wong CYB, Ononye S, Lopez DD, Engene N, McPhail KL, Gerwick WH & Balunas MJ (2013) Santacruzamate A, a potent and selective histone deacetylase inhibitor from the Panamanian marine cyanobacterium cf. *Symploca* sp. *J. Nat. Prod.* **76**: 2026–2033
- Romussi A, Cappa A, Vianello P, Brambillasca S, Cera MR, Dal Zuffo R, Fagà G, Fattori R, Moretti L, Trifirò P, Villa M, Vultaggio S, Cecatiello V, Pasqualato S, Dondio G, So CWE, Minucci S, Sartori L, Varasi M & Mercurio C (2020) Discovery of Reversible Inhibitors of KDM1A Efficacious in Acute Myeloid Leukemia Models. *ACS Med. Chem. Lett.* **11**: 754–759
- Saito A, Yamashita T, Mariko Y, Nosaka Y, Tsuchiya K, Ando T, Suzuki T, Tsuruo T & Nakanishi O (1999) A synthetic inhibitor of histone deacetylase, MS-27-275, with marked in vivo antitumor activity against

human tumors. *Proc. Natl. Acad. Sci. U. S. A.* **96**: 4592–4597

Schenk T, Chen WC, Göllner S, Howell L, Jin L, Hebestreit K, Klein H-U, Popescu AC, Burnett A, Mills K, Casero RA Jr, Marton L, Woster P, Minden MD, Dugas M, Wang JCY, Dick JE, Müller-Tidow C, Petrie K & Zelent A (2012) Inhibition of the LSD1 (KDM1A) demethylase reactivates the all-trans-retinoic acid differentiation pathway in acute myeloid leukemia. *Nat. Med.* **18**: 605–611

Sehrawat A, Gao L, Wang Y, Bankhead A 3rd, McWeeney SK, King CJ, Schwartzman J, Urrutia J, Bisson WH, Coleman DJ, Joshi SK, Kim D-H, Sampson DA, Weinmann S, Kallakury BVS, Berry DL, Haque R, Van Den Eeden SK, Sharma S, Bearss J, et al (2018) LSD1 activates a lethal prostate cancer gene network independently of its demethylase function. *Proc. Natl. Acad. Sci. U. S. A.* **115**: E4179–E4188

Sheng W, LaFleur MW, Nguyen TH, Chen S, Chakravarthy A, Conway JR, Li Y, Chen H, Yang H, Hsu P-H, Van Allen EM, Freeman GJ, De Carvalho DD, He HH, Sharpe AH & Shi Y (2018) LSD1 Ablation Stimulates Anti-tumor Immunity and Enables Checkpoint Blockade. *Cell* **174**: 549–563.e19

Somerville T, Salamero O, Montesinos P, Willekens C, Perez Simon JA, Pigneux A, Recher C, Popat R, Molinero C, Mascaro C, Maes T & Bosch F (2016) Safety, Pharmacokinetics (PK), Pharmacodynamics (PD) and Preliminary Activity in Acute Leukemia of Ory-1001, a First-in-Class Inhibitor of Lysine-Specific Histone Demethylase 1A (LSD1/KDM1A): Initial Results from a First-in-Human Phase 1 Study. *Blood* **128**: 4060–4060

Yang M, Gocke CB, Luo X, Borek D, Tomchick DR, Machius M, Otwinowski Z & Yu H (2006) Structural basis for CoREST-dependent demethylation of nucleosomes by the human LSD1 histone demethylase. *Mol. Cell* **23**: 377–387

25th Aug 2020

Dear Anna,

Thank you for the submission of your revised manuscript to EMBO Molecular Medicine. We have now received the enclosed reports from the two referees who reviewed the new version of your manuscript. As you will see, they are both supportive of publication, and I am thus pleased to inform you that we will be able to accept your manuscript pending the following final editorial amendments:

1) Main manuscript text:

- Please answer/correct the minor changes suggested by our data editors in the main manuscript file in track changes mode (document attached). Please use this file for any further modification.
- Thank you for providing "The Paper Explained" section. Please place it further down in the manuscript, above the references.
- Please modify the references format so as to have 10 authors before et al.
- In the material and method section, please indicate the source of cell lines and the gender and source of the mice.
- Thank you for providing the Data Availability section. Please move it to the end of the Material and Methods section. We note that the data are not yet publicly available, please note that they should be public before acceptance of the manuscript.
- Please remove the figures from the main manuscript, only the figure legends should remain.
- In the legends, please indicate exact p= values, not a range, along with the statistical test used.
- The references for Fig. 7H-I are missing in the main manuscript text (or mixed with Fig. 6H-I), please check.

2) Figures:

- We usually accommodate a maximum of 5 EV figures. Would you like to make one of your EV figures a main figure?
- The blots presented in Fig. 7C are very dark, would you have brighter versions?

3) Source Data:

Thank you for providing source data. Could you please upload one file per EV figure (containing PDF and excel file), similarly to what was done with the main figures?

4) You notified us that you agree with the publication of the Review Process File, thank you. Please let us know whether you also agree with the publication of the figures included in the point-by-point rebuttal letter.

The Authors checklist will be published at the end of the RPF.

I look forward to receiving your revised manuscript.

With my best wishes,

Lise

Lise Roth, PhD
Editor
EMBO Molecular Medicine

To submit your manuscript, please follow this link:

Link Not Available

The system will prompt you to fill in your funding and payment information. This will allow Wiley to send you a quote for the article processing charge (APC) in case of acceptance. This quote takes into account any reduction or fee waivers that you may be eligible for. Authors do not need to pay any fees before their manuscript is accepted and transferred to our publisher.

***** Reviewer's comments *****

Referee #1 (Remarks for Author):

The authors have addressed all concerns adding new data and discussing more in depth the study.

Referee #2 (Remarks for Author):

The authors adequately addressed all my previous questions. I recommend the paper for publication in EMBO mol med.

The authors performed the requested editorial changes.

8th Sep 2020

Dear Anna,

Thank you for submitting your revised version of the manuscript. I have now looked at everything and all is fine. I am therefore very pleased to accept your manuscript for publication in EMBO Molecular Medicine!

It will be sent to our publisher to be included in the next available issue of EMBO Molecular Medicine.

Please read below for additional important information regarding your article, its publication and the production process.

Congratulations on a very nice study!

With my best wishes

Lise

Lise Roth, Ph.D
Editor
EMBO Molecular Medicine

Follow us on Twitter @EmboMolMed
Sign up for eTOCs at embopress.org/alertsfeeds

*** ** IMPORTANT INFORMATION ** **

SPEED OF PUBLICATION

The journal aims for rapid publication of papers, using using the advance online publication "Early View" to expedite the process: A properly copy-edited and formatted version will be published as "Early View" after the proofs have been corrected. Please help the Editors and publisher avoid delays by providing e-mail address(es), telephone and fax numbers at which author(s) can be contacted.

Should you be planning a Press Release on your article, please get in contact with embomolmed@wiley.com as early as possible, in order to coordinate publication and release dates.

LICENSE AND PAYMENT:

All articles published in EMBO Molecular Medicine are fully open access: immediately and freely

available to read, download and share.

EMBO Molecular Medicine charges an article processing charge (APC) to cover the publication costs. You, as the corresponding author for this manuscript, should have already received a quote with the article processing fee separately. Please let us know in case this quote has not been received.

Once your article is at Wiley for editorial production you will receive an email from Wiley's Author Services system, which will ask you to log in and will present you with the publication license form for completion. Within the same system the publication fee can be paid by credit card, an invoice, pro forma invoice or purchase order can be requested.

Payment of the publication charge and the signed Open Access Agreement form must be received before the article can be published online.

PROOFS

You will receive the proofs by e-mail approximately 2 weeks after all relevant files have been sent to our Production Office. Please return them within 48 hours and if there should be any problems, please contact the production office at embopressproduction@wiley.com.

Please inform us if there is likely to be any difficulty in reaching you at the above address at that time. Failure to meet our deadlines may result in a delay of publication.

All further communications concerning your paper proofs should quote reference number EMM-2020-12525-V3 and be directed to the production office at embopressproduction@wiley.com.

Thank you,

Lise Roth, Ph.D
Scientific Editor
EMBO Molecular Medicine

Corresponding Author Name: Anna Obenaus
Journal Submitted to: EMBO Molecular Medicine
Manuscript Number: EMM-2020-12525